**Subject Category:**
Biology (whole organism)

cognition/ecology/behaviour

primates, foraging decisions, frugivory, food distribution, food preference

**Authors for correspondence:**
Cinzia Trapanese
e-mail: cinzia.trapanese@gmail.com
Benjamin Robira
e-mail: benjamin.robira@cefe.cnrs.fr

†Equal contribution.

# Where and what? Frugivory is associated with more efficient foraging in three semi-free ranging primate species

Cinzia Trapanese[1,2,3,4], Benjamin Robira[2,5],
Giordana Tonachella[2,6], Silvia di Gristina[2,6],
Hélène Meunier[3,4,†] and Shelly Masi[2,†]

[1]École Doctorale Frontières du Vivant, Centre de Recherches Interdisciplinaires, 8-8bis Rue Charles V, Paris, 75004, France
[2]UMR 7206 Éco-anthropologie (Muséum national d'Histoire naturelle-CNRS-Univ. Paris 7), Musée de l'Homme, 17 place Trocadéro, Paris, 75116, France
[3]Centre de Primatologie de l'Université de Strasbourg, Fort Foch, Niederhausbergen, 67207, France
[4]Laboratoire de Neurosciences Cognitives et Adaptatives, UMR 7364, CNRS et Université de Strasbourg, Strasbourg, 67000, France
[5]Institut de biologie de l'École normale supérieure (IBENS), École Normale Supérieure, CNRS, INSERM, PSL Research University, Paris, France
[6]Dipartimento di Scienze della Vita e Biologia dei Sistemi, Università degli Studi di Torino, Torino, Italy

 CT, 0000-0001-5396-8101; BR, 0000-0002-3168-6573

Foraging in seasonal environments can be cognitively challenging. Comparative studies have associated brain size with a frugivorous diet. We investigated how fruit distribution (*where*) and preference (*what*) affect foraging decisions in three semi-free ranging primate species with different degrees of frugivory: *Macaca tonkeana* ($N_{indiv} = 5$; $N_{trials} = 430$), *M. fascicularis* ($N_{indiv} = 3$; $N_{trials} = 168$) and *Sapajus apella* ($N_{indiv} = 6$; $N_{trials} = 288$). We used 36 boxes fixed on trees and filled with highly and less preferred fruits with different (weekly) spatio-temporal distributions. Individuals were tested in two conditions: (1) same fruit provided concurrently in the same quantity but in a scattered and in a clumped distribution, (2) highly preferred fruit was scattered while the less preferred was clumped. Generally, primates preferred feeding first on the boxes of the clumped distribution in both conditions, with the more frugivorous species at a higher degree than the less frugivorous species in condition (1), but not (2). Therefore, *what* fruit was available

changed the foraging decisions of the more frugivorous species who also engaged more in goal-directed travel. When feeding on preferred fruit, primates probably maximized foraging efficiency regardless of their degree of frugivory. Our findings emphasize that the food type and distribution may be a preponderant driver in cognitive evolution.

## 1. Introduction

Wild animals face decisive challenges to access vital resources. Difficulties involved in finding food vary depending on qualities such as a heterogeneous distribution and/or transient availability (e.g. African elephants [1]), a potentially demanding capture effort (e.g. goshawks [2]), and/or competitors feeding on the same resource (e.g. fish [3]; salamanders [4]). To maximize foraging efficiency, animals should attempt to increase energetic intake and minimize costs [5]. For frugivorous animals, food resources are patchy and variable in space and time, particularly in seasonal environments (e.g. bats [6]; primates [7]). In tropical forest characterized by fluctuating and annual patterns [8], locating ephemeral and scattered fruiting trees while avoiding competitors and predators can be energetically and cognitively demanding [9,10]. Nonetheless, a certain temporal periodicity in food production, hence location, may mean there is recurrent predictability over an individual's lifetime [11]. Consequently, the integration of salient cues with spatial information can be crucial to adjust travel and optimize foraging according to the quality (*what*), distribution and density (*where*) of the food available, particularly in seasonal environments (*when*).

Evolution of the ability to remember locations may be related to the distribution of resources and the foraging problems a species faces [12]. In this framework, the ability to use spatial memory to plan movements in advance and, thus, to forage efficiently, may be particularly profitable in more stable or predictably changing environments. A growing body of evidence shows that brain size and organization may relate to divergent ecological adaptations (*Ecological Intelligence Hypothesis*; [9,13]). In the last four decades, comparative studies on mammals have significantly associated brain size and the consequent high cognitive abilities to fruit richness in the diet, patchy fruit distribution and availability (mammals [14–16]; primates [17]). Generally, herbs and leaves are more homogeneously distributed and consistently present all year around, and hence more easily accessible than fruits [18,19]. With a diversity of diets within the taxon, primates are a good model for testing the *Ecological Intelligence Hypothesis* (e.g. [20,21]). For instance, differences in brain structure between the two gorilla species seem to be driven by divergent ecological habits [22,23]. Although not mutually exclusive, the idea that the environment is the core element in shaping the evolution of cognitive complexity has gained precedence over the view in which the social environment is the main overriding influence (the *Social Brain Hypothesis*, [24–27]).

Investigating the mechanisms underpinning foraging strategies in wild primates in terms of spatial cognition of food availability can be highly complex, as a number of environmental variables can act synergistically and change freely [28]. Nevertheless, it has been shown that wild capuchin monkeys are able to integrate memories of past feeding events (episodic memory), including *what*, *where* and *when* something happened [29]. Similarly, chimpanzees probably possess some abilities in classifying food trees, remembering quantity and frequency of fruit production across the year [30,31]. Such knowledge might then be used to maximize travel efficiency when planning to revisit certain trees [32]. Even in captivity, primates prefer selecting feeding locations that maximize the amount of food obtained and minimize the distance travelled (monkeys [33,34]; great apes [12]).

Studies in semi-free conditions allow the manipulation of variables while framing ecologically valid observations. We compared the individual spatial foraging strategies of three semi-free ranging primate species (*Macaca tonkeana*, *M. fascicularis* and *Sapajus apella*): while the main dietary tendency of the three species is frugivory, *M. tonkeana* are more frugivorous and tolerant [35,36] in comparison to *M. fascicularis*, which are frugivorous/omnivorous and despotic [36,37], and *S. apella*, which are even more omnivorous/insectivorous but socially quite tolerant [38,39] (electronic supplementary material, table S1). We investigated how primate foraging decisions were affected by clumped and scattered food distributions and food preference (highly versus less preferred, a proxy for food quality, e.g. [40]) by controlling and manipulating the variables *where* (food distribution) and *what* (fruit availability) in this semi-natural context. When the same fruit was found scattered versus clumped (*where*), we predicted that primates would feed first on the clumped fruit to increase foraging efficiency [5]. Since fruit trees provide ephemeral foods located in variable distributions, if fruit quality (in terms of preference) was

also a driving factor affecting their spatial foraging decisions, we predicted that when the most preferred fruit (*what*) was scattered, the more frugivorous primates, the *Macaca* spp. would take more into account both food quality and food distribution in comparison to the more generalist (omnivorous/insectivorous) species *S. apella* (electronic supplementary material, table S1), than when fruit quality did not vary between the two distributions. For the same reasons, the more frugivorous species were expected to rely more on the spatial memory of the food distribution and to have more goal-directed travel towards each feeding site in comparison to the more omnivorous species [41,42].

# 2. Material and methods

## 2.1. Subjects and food preference test

We studied three primate species living in social groups of 26 *Macaca tonkeana* (tested from June to October 2016), 17 *M. fascicularis* (July–October 2017) and 17 *S. apella* (May–September 2017), all captive-born. We tested 14 individuals in total: five *M. tonkeana*, three *M. fascicularis* and six *S. apella* (more details in electronic supplementary material, tables S2 and S3). The relatively low sample size per species (electronic supplementary material, table S3) is because we tested only individuals (i) with full development of cognitive capacities, i.e. adults (ii) who voluntary took part at the Habituation and the Test Phase, and (iii) who passed the criteria for the Habituation Phase (see section 'Habituation to the experimental protocol' in electronic supplementary material). Each group lived in a semi-controlled environment in vast wooded areas (1364–3788 m$^2$, electronic supplementary material, figure S1a,b,c) with natural vegetation at the Primate Centre of Strasbourg University, France. Primates had ad libitum access to water, and the regular daily provision of primate pellets. However, the supply of fruit (once per week) was suspended during the study period to increase subject motivation and to avoid excessive intake because of the experiments (see electronic supplementary material, 'Food preference test').

To select the preferred fruit used in these experiments, we first assessed a scale of fruit preference for each study group by presenting individuals with 10 sessions of 28 dichotomous choices among eight different appealing domestic fruit (see electronic supplementary material for details on the Food preference test). Since the three tested species are mainly frugivorous (range of the percentage of fruit in their diet, between 54–86%, see electronic supplementary material, table S1), fruit was an appetent food for all of them and no differences in the motivation to participate to the experiments were observed among the species.

## 2.2. Habituation to the experimental protocol

In the Test Phase, primates had to search for fruit inside 36 wooden boxes (25 × 17 × 18 cm) fixed on trees at 1 m above the ground in the outdoor area (electronic supplementary material, figures S2a,b and S3). We used a remote control system to unlock visited boxes from a distance, rendering the fruit available only when a focal subject [43] touched a box. This allowed us to test subjects individually within their social group. During the test, all boxes remained in the outdoor areas, but each week, only the boxes of the current task/season (a set of boxes in two different circular configurations, a clumped and a scattered one, details below) were baited and unlocked for the focal subjects. Each box was consistently filled with the same fruit provided in pieces in the same quantity (two pieces) and size (see electronic supplementary material for details). Therefore, the location of each box was associated solely to a given fruit type throughout the experiment. Non-appealing parts (fruit peels) of the available fruit in a given task/season (e.g. banana skin during the banana season), hereafter called *food cues*, were uniformly distributed each day across the outdoor area. Their function was (a) to discourage any box search via olfactory cues by standardizing the fruit olfactory environment, and (b) to intentionally provide the individuals with temporal cues for each season as it would happen in the wild.

Prior to the Test Phase, we carried out a Habituation Phase which consisted of two main steps. *The habituation in the indoor area* (i.e. experimental room) aimed at building the crucial associations that enabled us to individually test the subjects while keeping them in the social group: (a) a trial only started when the experimenter showed to the focal individual a *personal starting cue*, a specific object per individual (e.g. a plastic triangle, a felt-tip; electronic supplementary material, figure S4) that was presented with increasing delay over time (after 5 s at first, progressively increasing until 2.45 min) as a signal of the possibility to open the box for that subject, and (b) among the test boxes, the one containing fruit was the one corresponding to the *food cues* scattered on the ground (electronic

supplementary material, figure S5). The boxes containing fruit were unlocked with a remote control system only when the focal individual touched it. The individuals who reached the criterion (i.e. opening the baited box in three consecutive trials for each of the five fruits; see electronic supplementary material for details), passed to the second step of the habituation. *The habituation in the outdoor area* aimed to show the subjects the pattern of the spatio-temporal food availability of fruit of the Test Phase (see electronic supplementary material, 'Habituation to the experimental protocol' for more details). Two experimenters (CT, SG, or GT, and assistants) were in the outdoor area together with the study group. The subjects were introduced to the spatio-temporal availability of different fruits by adding the boxes of the specific fruit to the outdoor area week after week, following the task/season order in the Test Phase (see electronic supplementary material for more details). Thus, during the first week only the boxes of the first task/season were placed in the outdoor area. At the first day of the second week, the boxes of the second task/season were added to those of the first task/season which were no long available (i.e. neither baited nor openable). Progressively, also the boxes of the third and then of the fourth task/season were added in the same way. After the first set of four weeks (i.e. *The habituation in the outdoor area*) all boxes were always present in the outdoor area. Only individuals who successfully opened all baited boxes in at least two consecutive trials in each of the different task/seasons participated in the Test Phase (electronic supplementary material, table S3 for sample size of participating individuals).

## 2.3. Framework and experimental protocol of the Test Phase

The Test Phase was a sequence of four tasks, each lasting one week (from Monday to Friday) and repeated four times (16-weeks in total) with the same temporal order, mimicking the seasonal availability of fruit across different complete seasonal cycles (task/season 1, 2, 3, 4, 1, 2, 3, 4, etc.). In this study, we analysed the behaviour in three of the four tasks, the fourth task focusing on another topic and analysed elsewhere (Trapanese *et al.* [44]). Each subject was tested six to eight times per week (per task) by two experimenters (average weekly number of trials per species: *M. tonkeana* = 144 ± 1.53 trials, *M. fascicularis* = 56 ± 8.89, *S. apella* = 99 ± 9.3). To avoid giving them cues as to which boxes were available (baited) in a given trial, we pretended to fill also boxes of the other tasks/seasons (randomized at each trial) while filling the boxes of the current task/season (six more boxes were filled for *M. tonkeana* and 12 for *M. fascicularis* and *S. apella*; the difference is due to a protocol improvement after testing the *M. tonkeana*; see electronic supplementary material, 'Test Phase' for details). At each trial of a task/season the experimenter waited at the starting point for the passage of a possible focal individual. When a focal subject crossed the starting point, the experimenter showed to him/her his/her personal starting cue while keeping a neutral face expression and body posture (i.e. always orientated towards the centre of the outdoor area). This allowed us to avoid delivering unintentional cues to the focal individual about the position of the nearest baited box. When the focal individual moved, the experimenter followed him/her within 2 m (focal animal sampling [43]). Each time the individual stopped, the experimenter remained still, oriented towards the subject direction and waiting for the subject to move again (videos: https://www.dropbox.com/sh/ odwe7efw34dogef/AABs2KnjVR72c28NduQQNPYXa?dl=0). At each trial, the experimenter recorded on detailed maps of the enclosures: (1) the path used by the subject to visit the boxes, (2) the chronological order of the visited boxes (all numbered), (3) the identity of all visible individuals at each visited box and within 10 m of the starting point (figure 1*a*–*c*; more details below and in electronic supplementary material), and (4) all directional changes [43] of the focal individual due to the presence of a competitor (whose identity and hierarchical rank were noted) on the way to or in proximity of a box. Trials including directional changes due to competitors were excluded from the analyses ($N_{tot}$ = 18). We tried to minimize competition by having a second experimenter attracting the dominant individual(s) with food to a side corner of the area at the maximal distance from the boxes. The testing order of the subjects was randomized to minimize bias due to social learning (see electronic supplementary material, 'Test Phase').

## 2.4. *Where*: 'Clumped versus Scattered' Task (CvS 1 and 2), how does fruit distribution (clumped or scattered) affect primate foraging choices?

In the Clumped versus Scattered Task 1 (CvS1; $N_{trials\ tot}$ = 273) a highly preferred fruit from the food preference test (details on fruit choice per species in electronic supplementary material, tables S4–S6) was available in two circular distributions placed on opposite sides of the outdoor area (figure 1*a*,*b*). Each distribution was composed of six boxes, one with a clumped configuration (range distance between boxes 2–3 m) and the other with a more scattered arrangement (range distance between

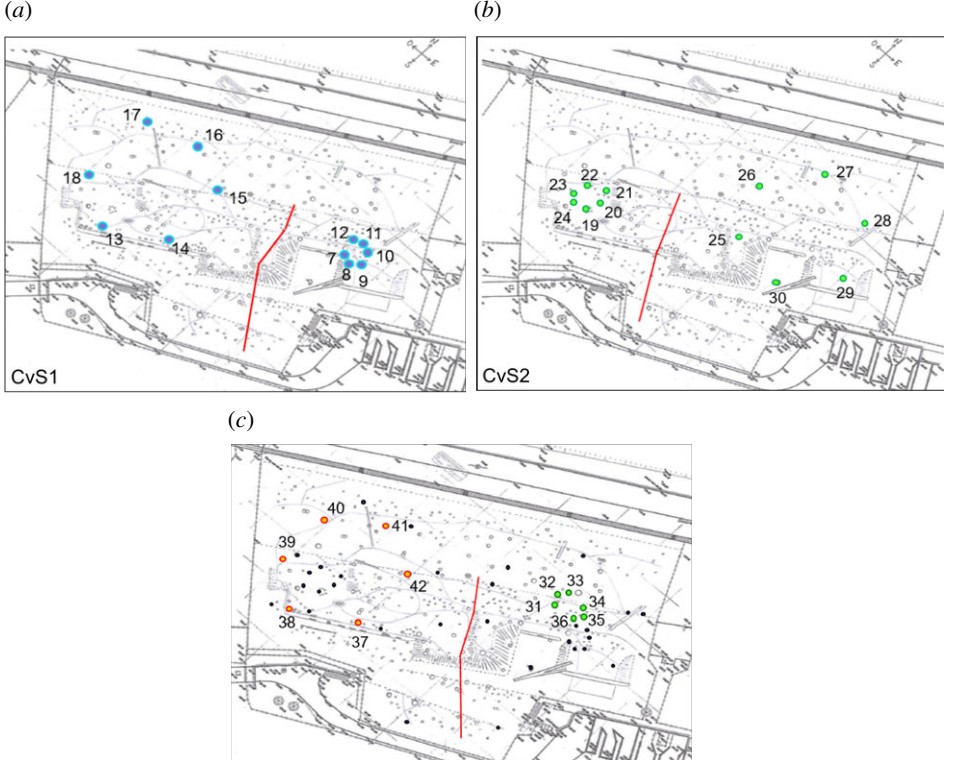

**Figure 1.** (*a*) Clumped versus Scattered Task 1 (CvS1). Position of the clumped and scattered distributions of the boxes in Clumped versus Scattered Task 1 in *M. tonkeana* outdoor area. The starting point of each trial was any point on the red line. (*b*) Clumped versus Scattered Task 2 (CvS2). Position of the clumped and scattered distributions of the boxes in Clumped versus Scattered Task 2 in *M. tonkeana* outdoor area. The starting point of each trial was any point on the red line. (*c*) Clumped versus Quality Task (CvQ). Position of the clumped and scattered distribution of the boxes in Clumped versus Quality Task in *M. tonkeana* outdoor area. The starting point of each trial was any point on the red line.

boxes 15–18 m). Because of the dense vegetation (see videos), the boxes of the same task/season were not visible from one another. The outdoor area was virtually split into two parts by a line whose points were equidistant from the closest box of each circular distribution (between 15 and 20 m) and served as starting point for the trials (see the red line in figure 1*a,b*). To check whether the choice of a distribution (clumped or scattered) was independent from the preference for a side of the outdoor area, we carried out a control task named CvS2 ($N_{\text{trials tot}} = 297$). In CvS2, the relative positions of the clumped and scattered distributions were inverted: if the clumped distribution was on the east side in CvS1, it was on the west side in CvS2 (figure 1*a,b*). The boxes were filled with a fruit similarly preferred to the one used in CvS1 (electronic supplementary material, table S6).

## 2.5. *Where* versus *what*: 'Clumped versus Quality' Task (CvQ), when primates forage on preferred fruit, does fruit quality play a more important role than fruit distribution?

This task/season also consisted of two circular distributions of six boxes disposed in the outdoor areas like in the CvS1 task ($N_{\text{trials tot}} = 316$; figure 1*c*). This time the six boxes of the scattered distribution were filled with the most preferred fruit from all individuals tested in the food preference test while the clumped distribution was concurrently baited with the least preferred fruit (electronic supplementary material, table S6 for fruit choice).

## 2.6. Statistical analysis

### 2.6.1. Assessing the preference for the clumped or scattered distribution: Index of Lateralization

To assess (a) if some individuals had a preference for a side of the enclosure independently from a preference for one of the fruit distributions (thus were unilateralized) and (b) the individual preference

for the clumped or scattered fruit distribution, we computed an Index of Lateralization. We did so adapting the Handedness Index HI $= (R - L)/(R + L)$ (e.g. [45]), with $R$ indicating the number of box choices in the right side of the outdoor area (i.e. the clumped distribution in CvS1 and the scattered one in CvS2) and $L$ indicating the number of choices in the left side of the outdoor area (i.e. the scattered distribution in CvS1 and the clumped one in CvS2; see figure 1a–c). We assessed the significance of the choice based on the Index of Lateralization using a binomial two-tailed test, to determine the lateralization degree of each individual ($N = 60$ minimum observations in one task/ season, see tables 2 and 3 for the results of the Index of Lateralization). We computed it on the first six boxes opened (as a circular resource distribution contained six boxes) in the CvS1 and CvS2 tasks/ seasons (with the two distributions inverted but containing the same fruit) *separately* for the purpose (a) (table 2), and *together* for the purposes (b) (table 3). If an individual showed a preference for the same side of the outdoor area in both CvS1 and CvS2 tasks/seasons, we assessed if the intensity of such preference (i.e. the rate of choice corresponding to a given side) varied between CvS1 and CvS2 using a proportional test (R function: prop.test). Individuals that were unilateralized (choosing consistently one side of the outdoor area regardless of the position of clumped or scattered distribution; $N = 4$; tables 2 and 3) were included in a subset of the statistical analyses (see section 'Models' description' for details) only if the intensity of their lateralization varied significantly (i.e. increased or decreased) from CvS1 to CvS2, thus showing a milder unilateralization.

### 2.6.2. Assessing reasons for a preference for a fruit distribution and the strategies used to forage: generalized linear mixed models

In order to compare species responses to the different foraging conditions, using R software version 3.4.3 (R Development Core Team, 2017), we ran generalized linear mixed models (GLMMs; [46]) with binomial error structure and logit link function [47] as the output considered was whether the behaviour of interest was observed or not (e.g. box visited from the clumped distribution or not; change of the distribution or not; box baited or not; for the model formulae see table 1). Specifically, the linear models aimed to complement the analysis of the Index of Lateralization by better qualifying and quantifying the strategy that was used. Namely, we constructed four types of models that we ran on subsets of our data considering specific tasks/seasons at a time.

With a first model ('First visited box') we assessed whether the decision to forage was made initially by investigating whether the first visited box belonged to the clumped distribution or not. This allowed us to assess if individuals had memorized the spatio-temporal food availability, targeting at first a box of the clumped distribution (to forage efficiently; see electronic supplementary material for further rational). To disentangle the effect of the distribution (*where*) and the quality of fruit (*what*) on such a choice, we conducted the analysis separately: first on CvS1 and CvS2 tasks/seasons ($N_{\text{visited boxes}} = 508$), and then on CvS1 and CvQ tasks/seasons ($N_{\text{visited boxes}} = 478$) in which the distributions were identically distributed in the outdoor area, but fruit quality differed (figure 1a–c; electronic supplementary material, table S6).

Second, we ran another model ('Successive five boxes visited') to characterize the foraging strategy (discussed in electronic supplementary material, 'Models' goal'). For instance, individuals could target the closest neighbouring box, or rather prefer to leave for the opposite distribution after having fed on a given box. To depict this conditionality, we included in the model only data from the second visited box to the sixth one, but included the 'location of the previous box' (i.e. from the clumped or scattered distribution) as a tested variable (i.e. first order conditionality, see models' description and formula in table 1). We considered only till the sixth visited box since each circular distribution contained only six boxes and we assumed that an omniscient individual would forage consecutively on the six boxes from the clumped distribution. Once again, this model was conducted on the two CvS tasks/seasons together (CvS1 and CvS2; $N_{\text{visited boxes}} = 2466$), and separately on the dataset from CvS1 and CvQ pooled together ($N_{\text{visited boxes}} = 2841$).

Third, in the model 'Change of the distribution' ($N_{\text{obs}} = 4281$), we assessed why subjects tended to leave the distribution (i.e. clumped or scattered one) they were foraging in (within the series of the first six boxes). Specifically, we tested for the propensity to stay in a given distribution or to leave it depending on (1) whether the previously visited box was baited or not, (2) its location, whether this box belonged to the clumped or the scattered distribution, (3) the tested species, (4) the task/season and (5) the rank of the focal subject. As before, we assessed the choice of the box after the first one, from the second visited box to the sixth one (see the reasons mentioned above; table 1). Last, to assess whether individuals developed a fine-scale spatial memory or not, we investigated whether the

**Table 1.** Influence of the descriptive variables on the probability to choose the clumped distribution, to change the distribution or the baited box of the current task/season; results of the full models (GLMMs). est. = estimate, s.d. = standard deviation, d.f. = degree of freedom, $\chi^2$ = statistics value, $N$ = sample size. Estimate and standard deviation always refer to the difference between the reported level and the corresponding reference category (i.e. included in the intercept). In the model formula, bold variables refer to tested variables. $(1 + <\text{variable } n> | <\text{random effect}>)$ indicates the presence of a random effect on the intercept (1) or the slope of a given variable (variable $n$), intercept and slope being correlated. Bold variables are the significant ones after correction for maintaining the FDR to 0.05. Dummy-coded variables are labelled with the suffix 'dummy'. Models 'First visited box CvS' and 'First visited box CvQ' did not differ significantly from their respective null model, hence they were not further investigated, and the individual effect of each variable is therefore not displayed in this table.

| | est. | s.d. | d.f. | $\chi^2$ | p-value |
|---|---|---|---|---|---|
| model 'successive five visited boxes CvS' ($N = 2466$): | choice of the clumped distribution (0 = no, 1 = yes) ~ **Task + Species + Rank + No. of competitors at the box + Distribution of previous box** + Day time + Session + (1 + **No. of competitors at the box + Distribution of previous box** + Session\|Subject) | | | | |
| intercept[b] | −2.207 | 0.405 | a | a | a |
| task (CvS2) | −0.156 | 0.265 | 1 | 0.329 | 0.566 |
| species (*M. fascicularis*) | 0.400 | 0.333 | 2 | 9.431 | **0.009** |
| species (*M. tonkeana*) | 1.273 | 0.275 | | | |
| rank (subordinate) | 0.096 | 0.257 | 1 | 0.066 | 0.797 |
| No. of competitors at the box[c] | −0.210 | 0.102 | 1 | 2.762 | 0.096 |
| distribution of previous box (clumped) | 4.334 | 0.295 | 1 | 35.878 | **<0.001** |
| day time (morning) | −0.076 | 0.165 | 1 | 0.207 | 0.649 |
| session[d] | 0.152 | 0.093 | 1 | 2.024 | 0.155 |
| model 'successive five visited boxes CvQ' ($N = 2841$): | choice of the clumped distribution (0 = no, 1 = yes) ~ **Task + Species + Rank + No. of competitors at the box + Distribution of previous box** + Day time + Session + (1 + **Task + No. of competitors at the box + Distribution of previous box** + Session\|Subject) | | | | |
| intercept[e] | −2.366 | 0.469 | a | a | a |
| task (CvQ) | −0.360 | 0.233 | 1 | 2.243 | 0.134 |
| species (*M. fascicularis*) | 0.523 | 0.331 | 1 | 1.760 | 0.185 |
| species (*M. tonkeana*) | 0.489 | 0.421 | 2 | 3.200 | 0.202 |
| rank (subordinate) | 1.283 | 0.358 | 1 | 1.189 | 0.276 |

*(Continued.)*

**Table 1.** (Continued.)

| | est. | s.d. | $\chi^2$ | d.f. | p-value |
|---|---|---|---|---|---|
| No. of competitors at the box[f] | 0.024 | 0.137 | 0.026 | 1 | 0.873 |
| distribution of previous box (clumped) | 4.638 | 0.414 | 27.613 | 1 | **<0.001** |
| day time (morning) | −0.128 | 0.185 | 0.450 | 1 | 0.502 |
| session[g] | 0.051 | 0.091 | 0.253 | 1 | 0.615 |
| model 'change of the distribution' (N = 4281): change distribution (0 = no, 1 = yes) ~ **Task + Species + Rank + Nature of previous box (baited versus Non-baited) + Distribution of previous box** + Day time + Session + (1 + **Task + Nature of previous box + Distribution of previous box** + Session\|Subject) | | | | | |
| intercept[h] | −2.075 | 0.379 | a | a | a |
| task (CvS2)* | 0.131 | 0.255 | 7.797 | 2 | 0.020 |
| task (CvQ)* | −0.483 | 0.185 | | | |
| species (M. fascicularis) | −0.259 | 0.222 | 2.837 | 2 | 0.242 |
| species (M. tonkeana) | −0.463 | 0.185 | | | |
| rank (subordinate) | 0.276 | 0.158 | 2.154 | 1 | 0.142 |
| nature of previous box (non-baited) | 1.453 | 0.321 | 10.604 | 1 | **0.001** |
| distribution of previous box (clumped)* | −0.740 | 0.441 | 5.710 | 1 | 0.017 |
| day time (morning) | 0.136 | 0.134 | 0.992 | 1 | 0.319 |
| session[i] | 0.077 | 0.077 | 1.058 | 1 | 0.304 |
| model 'goal-directed strategy' (N = 5167): non-baited box (0 = no, 1 = yes) ~ **Species + Task * Distribution of box + Rank** + Day of time + Session + (1 + **task_dummy_1:circle_box_dummy + task_dummy_2:circle_box_dummy**\|Subject) + (1 + **task_dummy_1 + task_dummy_2**\|Subject) + (1 + **circle_box_dummy**\|Subject) + (1 + Session\|subject) | | | | | |
| intercept[j] | −1.362 | 0.492 | a | a | a |
| species (M. fascicularis) | −1.116 | 0.420 | 21.469 | 2 | **<0.001** |
| species (M. tonkeana) | −2.945 | 0.435 | | | |

(Continued.)

**Table 1.** (*Continued.*)

| | est. | s.d. | $\chi^2$ | d.f. | *p*-value |
|---|---|---|---|---|---|
| task (CvS2) | −1.598 | 0.326 | a | a | a |
| task (CvQ) | 0.069 | 0.336 | a | a | a |
| distribution of box (clumped) | −0.983 | 0.305 | a | a | a |
| task (CvS2):distribution of box (clumped) | 2.001 | 0.452 | 20.543 | 2 | **<0.001** |
| task (CvQ):distribution of box (dumped) | −1.045 | 0.471 | | | |
| rank (subordinate) | −0.486 | 0.332 | 1.316 | 1 | 0.251 |
| day time (morning) | −0.037 | 0.192 | 0.037 | 1 | 0.848 |
| session[k] | −0.361 | 0.087 | 9.137 | 1 | **0.003** |

[a]Not shown because having a limited interpretation.

[b]Intercept refers to the reference level, here, dominant *S. apella* in task CvS1 during the afternoon when they foraged on the scattered distribution previously.

[c]z-transformed to a mean of zero and a standard deviation of one. Mean and standard deviation before z-transformation were respectively 0.922 and 1.183.

[d]z-transformed to a mean of zero and a standard deviation of one. Mean and standard deviation before z-transformation were respectively 3.132 and 1.163.

[e]Intercept refers to the reference level, here, dominant *S. apella* in task CvS1 during the afternoon when they foraged on the scattered distribution previously.

[f]z-transformed to a mean of zero and a standard deviation of one. Mean and standard deviation before z-transformation were respectively 0.855 and 1.065.

[g]z-transformed to a mean of zero and a standard deviation of one. Mean and standard deviation before z-transformation were respectively 3.048 and 1.179.

[h]Intercept refers to the reference level, here, dominant *S. apella* in task CvS1 during the afternoon when they foraged on the scattered distribution previously and opened a baited box.

[i]z-transformed to a mean of zero and a standard deviation of one. Mean and standard deviation before z-transformation were respectively 3.016 and 1.190.

[j]Intercept refers to the reference level, here, dominant *S. apella* during the afternoon when they foraged on the scattered distribution in task CvS1.

[k]z-transformed to a mean of zero and a standard deviation of one. Mean and standard deviation before z-transformation were respectively 3.016 and 1.189.

*Not significant when keeping the false discovering rate constant (see Material and methods). New threshold of significance is 0.001.

**Table 2.** Measures of laterality for each subject for each task/season. rank sub = subordinate, rank dom = dominant, RC CvS1 = equivalent of the Right Handedness Index, Clumped versus Scattered Task 1 (CvS1); RC CvS2 = equivalent of the Right Handedness Index, Clumped versus Scattered Task 2 (CvS2); $p$ = $p$-value; $z$ = $z$-score; lat = laterality; obs = number of observations.

| subject | species | rank | RC CvS1 | RC CvS1 $p$ | RC CvS1 z-score | RC CvS1 lat | RC CvS1 obs | RC CvS2 | RC CvS2 $p$ | RC CvS2 z-score | RC CvS2 lat | RC CvS2 obs |
|---|---|---|---|---|---|---|---|---|---|---|---|---|
| Laeticiette | M. fascicularis | sub | 0.292 | 0.006 | 2.858 | right side (clumped) | 96 | 0.528 | <0.001 | 4.478 | right side (scattered) | 72 |
| Tempete | M. fascicularis | dom | −0.129 | 0.278 | −1.193 | none | 85 | −0.423 | <0.001 | −4.163 | left side (clumped) | 97 |
| Nicolette | M. fascicularis | sub | 0.797 | <0.001 | 8.653 | right side (clumped) | 118 | 0.346 | <0.001 | 3.577 | right side (scattered) | 107 |
| Willow | S. apella | dom | −0.326 | 0.002 | −3.181 | left side (clumped) | 95 | −0.333 | 0.001 | −3.416 | left side (scattered) | 105 |
| Franklin | S. apella | sub | 0.373 | 0.001 | 3.403 | right side (scattered) | 83 | −0.852 | 0.000 | −8.853 | left side (scattered) | 108 |
| Koli | S. apella | sub | −0.361 | 0.003 | −3.064 | left side (clumped) | 72 | 0.030 | 0.902 | 0.246 | none | 66 |
| Litchi | S. apella | sub | −0.100 | 0.519 | −0.775 | none | 60 | 0.133 | 0.188 | 1.411 | none | 113 |
| Popeye | S. apella | dom | −0.139 | 0.289 | −1.179 | none | 72 | −0.121 | 0.246 | −1.257 | none | 107 |
| Kolette | S. apella | sub | −0.357 | 0.001 | −3.273 | left side (clumped) | 84 | 0.412 | <0.001 | 4.159 | right side (clumped) | 102 |
| Olli | M. tonkeana | sub | 0.204 | 0.005 | 2.857 | right side (clumped) | 196 | −0.449 | <0.001 | −6.562 | left side (clumped) | 214 |
| Yannick | M. tonkeana | dom | −0.253 | <0.001 | −3.553 | left side (scattered) | 198 | −0.479 | <0.001 | −7.322 | left side (clumped) | 234 |
| Walt | M. tonkeana | sub | 0.800 | <0.001 | 9.798 | right side (clumped) | 150 | −0.866 | <0.001 | −10.568 | left side (clumped) | 149 |
| Wallace | M. tonkeana | dom | 0.530 | <0.001 | 6.472 | right side (clumped) | 149 | −0.567 | <0.001 | −6.565 | left side (clumped) | 134 |
| Nereis | M. tonkeana | dom | 0.306 | <0.001 | 3.667 | right side (clumped) | 144 | −0.379 | <0.001 | −4.352 | left side (clumped) | 132 |

**Table 3.** Measures of laterality for each subject considering task/season CvS1 and CvS2 pooled together. rank sub = subordinate, rank dom = dominant.

| subject | species | rank | clumped choice | $p$-value | Z-score | laterality | obs |
|---|---|---|---|---|---|---|---|
| Laeticiette | M. fascicularis | sub | $-0.060$ | $4.88 \times 10^{-1}$ | $-0.772$ | none | 168 |
| Tempete | M. fascicularis | dom | 0.165 | $3.13 \times 10^{-2}$ | 2.224 | clumped | 182 |
| Nicolette | M. fascicularis | sub | 0.253 | $1.76 \times 10^{-4}$ | 3.800 | clumped | 225 |
| Willow | S. apella | dom | $-0.02$ | $8.32 \times 10^{-1}$ | $-0.283$ | none | 200 |
| Franklin | S. apella | sub | $-0.644$ | $4.32 \times 10^{-20}$ | $-8.899$ | scattered | 191 |
| Koli | S. apella | sub | 0.203 | $2.12 \times 10^{-2}$ | 2.384 | clumped | 138 |
| Litchi | S. apella | sub | 0.121 | $1.28 \times 10^{-1}$ | 1.510 | none | 173 |
| Popeye | S. apella | dom | $-0.017$ | $8.81 \times 10^{-1}$ | $-0.242$ | none | 179 |
| Kolette | S. apella | sub | 0.387 | $1.36 \times 10^{-7}$ | 5.279 | clumped | 186 |
| Olli | M. tonkeana | sub | 0.332 | $1.71 \times 10^{-11}$ | 6.717 | clumped | 410 |
| Yannick | M. tonkeana | dom | 0.144 | $3.29 \times 10^{-3}$ | 2.983 | clumped | 432 |
| Walt | M. tonkeana | sub | 0.833 | $3.87 \times 10^{-54}$ | 14.400 | clumped | 299 |
| Wallace | M. tonkeana | dom | 0.548 | $5.17 \times 10^{-21}$ | 9.214 | clumped | 283 |
| Nereis | M. tonkeana | dom | 0.341 | $1.59 \times 10^{-8}$ | 5.658 | clumped | 276 |

visited box was baited or not (model 'Goal-directed strategy'). Again, for the same reasons mentioned above, we only considered the first six boxes. In this case the model was run on the dataset of all three tasks together considering data from the first visited box to the sixth one (CvS1, CvS2, CvQ; $N_{\text{visited boxes}} = 5167$).

## 2.7. Models' description

Since feeding motivation might vary throughout the day, we included in all models the time of the day (morning or afternoon) as a control predictor. To control for learning over time, we added the session number as an ordinate control predictor (table 1).

In 'First visited box CvS' model (that considers the distribution, clumped versus scattered, to which the first box visited by the focal individual was the closest to), we included task/season as a control variable, to control for the position of the distributions in the outdoor area given that although individuals showing robust unilateralization were excluded from the analysis ($N = 2$ individuals, see §3.1), the individuals with milder unilateralization (thus having a decrease in the intensity of their lateralization from CvS1 to CvS2) were included in the analysis. Additionally, we tested the variable species since they might differ in their spatial cognitive abilities. Tested species also differ in their degree of despotism [48,49], which could potentially affect their foraging decision choices. To determine if, and how, these variables affected foraging decisions, we tested for the interaction between species, rank and number of competitors present at the starting point (in a circular zone of radius 10 m around it; see table 1).

In the 'Successive five boxes visited CvS' model (which assesses how robust the choice of a distribution was after the first box and the strategy used to forage) the same variables were tested. Note that in this model, however, we used the number of competitors in a circular zone of radius 10 m around the chosen box in place of the number of competitors at the starting point. We also added the distribution of the previous visited box. As the influence of the distribution might be relative to the species, as they might differ in their foraging strategy, distribution of the previous box was tested in interaction with species (table 1). We similarly constructed the models 'First visited box CvQ' and 'Successive five visited boxes CvQ' (see formula in table 1). Since one individual (Nereis) of M. tonkeana did not match group fruit preference for the CvQ task/season, hence not fitting the experimental settings, we removed her for the two models testing for a distribution preference in relation to the fruit preference (*Where* versus *what*: 'First visited box CvQ' and 'Successive five visited boxes CvQ'; see electronic supplementary material for more details on model stability investigation

and the exclusion of this individual from a part of the analysis). It is worth noticing that, for those later four models ('First visited box CvS or CvQ', and 'Successive five visited boxes CvS or CvQ'), we focused on the first six boxes visited regardless their contents, i.e. if they were baited or empty, but considering their location: for baited boxes, we considered if they were located in the clumped or in the scattered distribution; for empty boxes, the proximity to baited boxes of one of the two distributions was considered (i.e. considering the line of the starting point, if they were located in the side of the clumped or scattered distribution for each task/season).

In the model 'Change of the distribution' we hypothesized that the probability of leaving when discovering a non-baited box may be dependent on the current location (distribution: clumped or scattered); therefore, whether the previous box is baited or not and its location were initially considered in interaction. We also tested whether each species was equally susceptible to leave for the other distribution depending on the task/season. Thus, species and task were initially considered in interaction. We also included the rank of the focal individual as a testing variable as the individual's strategy might be conditional to the social position within the group (table 1).

Finally, we ran the model 'Goal-directed strategy' on the first six baited boxes (of each task/season) for all tasks/seasons together (CvS1, CvS2, CvQ) given that this is related to all foraging activities. Here, we investigated whether the box opened was baited or not. In such a framework, individuals unilateralized that were excluded from the other models could here be included in the sample, as the goal was no longer to assess the choice of the side of the distribution they forage in, but how accurate they were when visiting a box. For such a behaviour, we therefore no longer discarded unilateralized individuals, or individuals that did not have the fruit preference matching the group preference, as long as we tested for the descriptive variables of those features (i.e. task/season and distribution of the box), while accounting for individual differences with a random factor. Indeed, as previously, we assumed that species' foraging strategy might differ, particularly with regard to the fruit type provided, hence we started with an interaction between species and task/season. We also assumed that the distribution individuals are foraging in, relatively to the fruit type (hence, relatively to task/season), could be influential, therefore we considered these terms in interaction. Finally, since individual spatial skills might stem from their relative social position (i.e. dominant versus subordinate), we tested for rank (table 1).

To maintain the type I error at or below the nominal level of 5% [50,51] for all models we also included subject as a random effect separately on the intercept and the slope of all necessary variables but time of the day, to limit model complexity such as this was only a control variable and we hypothesized weak inter-individual differences. We therefore did so for competitors ($z$-transformed, model 'First visited box' and 'Successive five visited boxes'), previous box distribution ('Successive five visited boxes'), current box distribution ('Goal-directed strategy'), session and context (task/season). If we associated a random effect to the control variable, we also controlled for the random effect. If a variable was included in an interaction, we generally included a random effect with regard to this interaction. Due to computational limits, if random effects were too numerous, we excluded correlations among random effects on different predictors and dummy-coded categorical predictors (table 1). Results displayed are only the latest version of models obtained after sequentially removing non-significant terms of higher complexity; it handles from correlations terms (to lower risks of overparametrization and misinterpretation, discussed in [52] and electronic supplementary material, 'Model selection'). More details on the implementation of the models are in the electronic supplementary material.

Last, we used a two-tailed binomial test to assess whether the probability of visiting at least three baited boxes (electronic supplementary material for reason of the choice) of the scattered distribution after visiting all boxes of the clumped distribution was above chance level.

## 2.8. Models implementation

GLMMs were fitted using the glmer function from the lme4 package [53] while variance inflation factor (VIF; [54]) assessments were conducted using the function vif from the car package [55]. Correlation, assessed on equivalent models not including random effects (i.e. GLM), was not an issue ($VIF_{max} < 2$ among all models, that maximum being reached for 'Goal-directed strategy' model, see electronic supplementary material for investigations on this specific case; for the other models it ranges below 2). To ease the interpretability of the results by allowing to have comparable estimates [56] we scaled all the continuous predictors. Only the number of competitors was left skewed due to zero inflation, but no transformation allowed reaching a better symmetrical distribution, except doing only a binomial variable, being null if no competitor, or one if at least one competitor was present. Doing so would have, though, greatly impeded interpretations and we therefore kept the variables as

integers. To assess, however, the effect of potential leverage points corresponding to the right 'queue' of this variable, we ran sub-models considering only values having at least 20 observations. Significance of results remained unchanged, therefore we present the results considering all cases. We investigated models' predictions sensibility using various statistical indicators and sub-models. Overall, models did not seem to suffer from the reduced sample, but interpretations must nonetheless remain cautious (further details in electronic supplementary material).

To determine if the tested variables had an effect together (i.e. if adding our tested variables in the model improved the fit of the data), the 'full' model, containing all predictors, was compared to the 'null' model [57], containing only the control variables and the necessary random effects, using a likelihood ratio test (R-function anova set to 'Chisq', [58]). To evaluate the singular effect of each variable, we discarded the corresponding variable only, and compared the deviance of this reduced model with the one of the full models (R-function drop1; [51]). We maintained the false discovery rate at the nominal level of 0.05 for each analysis that we conducted containing at least five independent tests, by applying Benjamini and Liu's second procedure [59]. That is, we did so for the lateralization investigation, taking into account $p$-values obtained with the binomial test, for each generalized linear mixed model, using the $p$-values obtained with the drop1 function, and finally, for the pairwise comparisons depicted in each figure. Whenever necessary, we refer to the corrected threshold of significance.

# 3. Results

## 3.1. Assessment of strong and mild unilateralized individuals using the Index of Lateralization

Among all 14 tested individuals, four had an Index of Lateralization indicating a preference for the same side of the outdoor area in both CvS tasks/seasons (*M. tonkeana*: $N = 1$, *M. fascicularis*: $N = 2$, *S. apella*: $N = 1$, table 2). Two individuals (a subordinate from *M. fascicularis* and a dominant from *S. apella*) did not vary in the intensity of their side preference (see Material and methods), thus they were excluded from the analyses of the models 'First visited box' and 'Successive five visited boxes' where the side preference was assessed, but not from the models 'Change of the distribution' and 'Goal-directed strategy' (see reasons in 'Models' description'). On the contrary, the other two individuals presented a significant change in the intensity of this preference between the two tasks, thus their preference for one side of the area was considered mild (see §2.6.1).

## 3.2. *Where*: 'Clumped versus Scattered distribution'

When considering the Index of Lateralization for both CvS tasks/seasons together to check for the preference for the clumped or scattered distribution (table 3), the two individuals with mild lateralization significantly (after Benjamini and Liu's correction) preferred the clumped distribution (see electronic supplementary material), therefore demonstrating a significant reinforcement of their bias when it matched the location of the clumped distribution. Among the 10 individuals (four *M. tonkeana*, five *S. apella* and one *M. fascicularis*) that exhibited variation in side preference in the two CvS tasks/ seasons, all four *M. tonkeana* significantly preferred foraging on the clumped distribution (see Index of Lateralization in table 3). Among the five *S. apella*, one individual significantly preferred foraging on the clumped distribution (Kolette), another one significantly preferred foraging on the scattered distribution (Franklin, subordinate), two individuals did not show preference in any of the two CvS tasks/seasons (Litchi and Popeye), and the last one showed preference for the clumped distribution (Koli, table 3; but her preference was only in the task/season CvS1 if the index was computed separately in the two tasks/seasons, see table 2). One individual of *M. fascicularis* showed preference for the clumped distribution (Tempete, table 3; but the preference was only in one task/season in CvS2 when the index was computed separately in the two tasks/seasons, see table 2). In sum, 75% of non-unilateralized individuals ($N = 12$) preferred the clumped distribution based on the Index of Lateralization.

We then tested for the preference for one distribution based on solely the first box chosen using GLMMs. The probability to choose a box of the clumped distribution first was not significant nor influenced by the social environment (i.e. number of competitors) or the individual features (e.g. rank, species; full/null model comparison: model 'First visited box CvS': $\chi^2 = 2.194$, d.f. $= 8$, $p = 0.975$). However, the model 'Successive five visited boxes CvS' significantly differed from the corresponding null model (full/null model comparison: $\chi^2 = 1371.3$, d.f. $= 14$, $p < 0.001$; table 1). The probability of selecting a box from the clumped distribution was higher if the previously visited box was also from

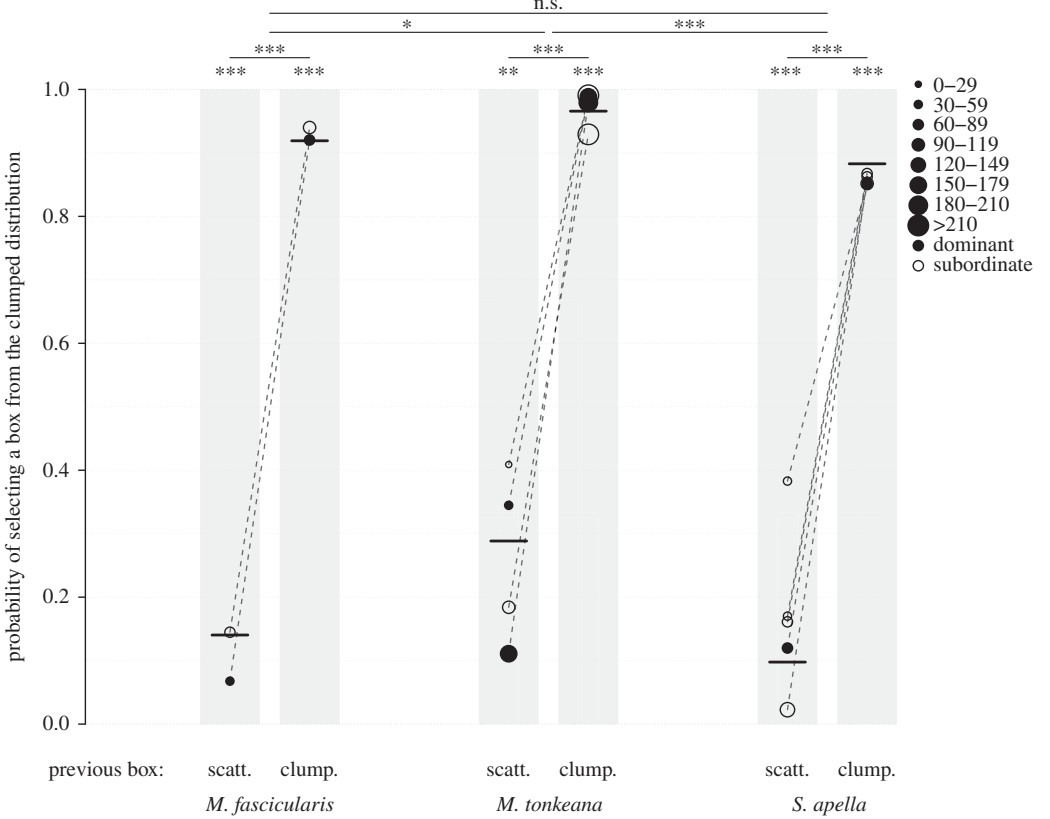

**Figure 2.** Influence of the species and the location of the previous box chosen on the probability of selecting a box from the clumped distribution (model 'Five successive visited boxes CvS'). scatt. = previous box selected belonged to the scattered distribution, clump. = previous box selected belonged to clumped distribution. The lines show the difference among species (not dependent on the previous box since the interaction was not significant). In figure 2 but also 3 and 4, black lines indicate the fitted model (table 1) with non-plotted categorical predictors manually dummy-coded and scaled, and non-plotted continuous predictor scaled, to have a null effect here. Raw data might include the effect of those variables, hence visual difference with the model. Dotted lines indicate paired observations. Each circle represents one individual and the size of each represents the number of total actions scored. Depicted significance indicates comparison to neutral choice (i.e. probability of 0.5) or between two categories (i.e. between categories pointed out by the end of the horizontal plain line). To control for the false discovery rate (FDR), we used Benjamini and Liu's procedure [59]. The significance of the test is therefore related to the corrected threshold ($\alpha'$) to keep the FDR at the nominal level of 0.05, and is depicted as following: n.s. = not significant, *: $p$-value $\leq \alpha'$, **: $p$-value $< \alpha'/10$, ***: $p$-value $< \alpha'/100$.

the clumped distribution (estimates $\pm$ s.d. = 4.334 $\pm$ 0.295, $\chi^2$ = 35.878, d.f. = 1, $p <$ 0.001, figure 2). Species differences existed, with *M. Tonkeana* having a significantly stronger probability to select a box from the clumped distribution (species differences: $\chi^2$ = 9.431, d.f. = 2, $p$ = 0.009; *M. tonkeana* estimate $\pm$ s.d.= 1.273 $\pm$ 0.275, reference is *S. apella*, table 1). No influence of the task ($\chi^2$ = 0.329, d.f. = 1, $p$ = 0.566), rank ($\chi^2$ = 0.066, d.f. = 1, $p$ = 0.797), number of competitors at the box ($\chi^2$ = 2.762, d.f. = 1, $p$ = 0.096), learning (session: $\chi^2$ = 2.024, d.f. = 1, $p$ = 0.155) or time of day ($\chi^2$ = 0.207, d.f. = 1, $p$ = 0.649; table 1) were, however, found.

## 3.3. Where versus what: clumped versus quality (preferred fruit)

As previously, in this condition in which the most preferred food was in the scattered distribution, the probability to choose the clumped distribution with the first box visited was not significant, nor was it influenced by the number of competitors, rank or species (full/null model comparison of 'First visited box CvQ': $\chi^2$ = 3.062, d.f. = 12, $p$ = 0.995). Model 'Successive five visited boxes CvQ' significantly differed from the corresponding null model (full/null model comparison: $\chi^2$ = 1437.700, d.f. = 18, $p <$ 0.001). The probability of selecting the clumped distribution was higher if the previously visited box was also from the clumped distribution (estimates $\pm$ s.d.= 4.638 $\pm$ 0.414, reference is the scattered distribution, table 1). It is worth noticing that the intensity of estimates between this model

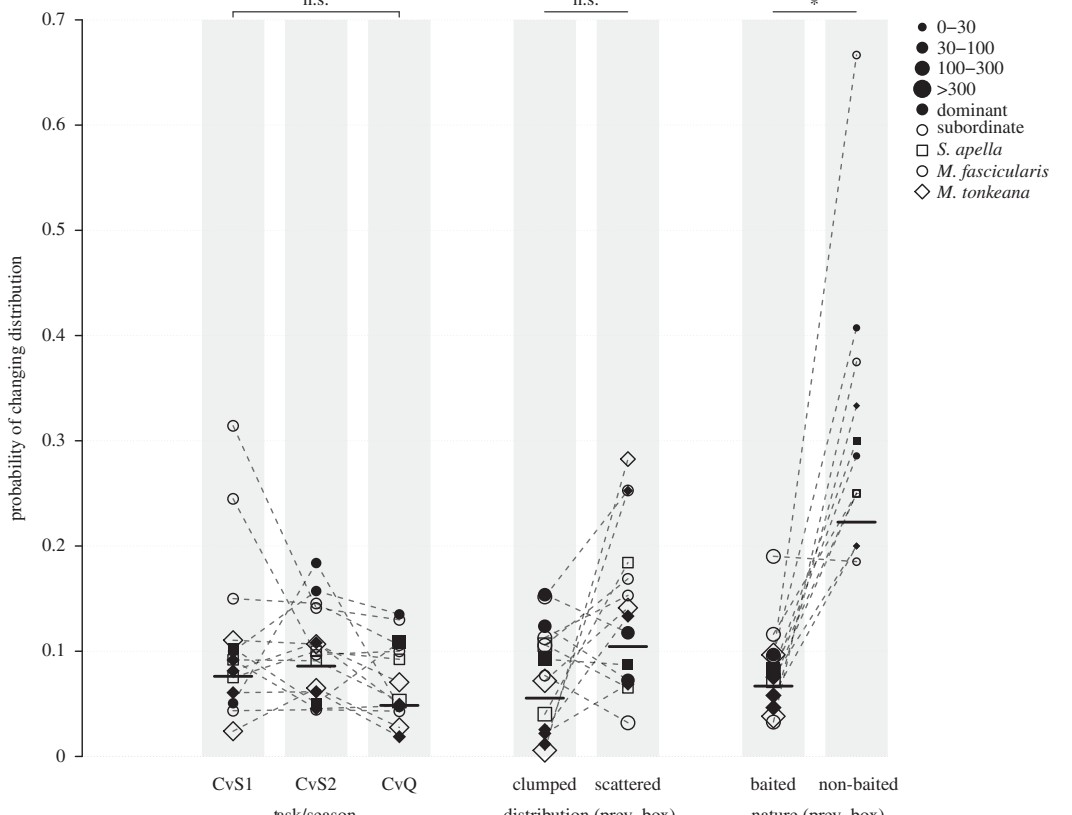

**Figure 3.** Influence of the task/season, the location of the previous target box (clumped or scattered distribution) and of the nature of the previous box (baited or not) on the probability of switching of targeted distribution (model 'Change of the distribution'). CvS1 = Clumped versus Scattered Task 1, CvS2 = Clumped versus Scattered Task 2, CvQ = Clumped versus Quality Task. See legend of figure 2 for further information.

and the model 'Successive five visited boxes CvS' previously described does not greatly differ. Although species had no effect in the presented model ($\chi^2 = 3.200$, d.f. = 2, $p = 0.202$), the analysis revealed potentially significant effect of species in paired interaction with task/season ($\chi^2 = 7.155$, d.f. = 2, $p = 0.028$), the number of competitors at the box ($\chi^2 = 9.096$, d.f. = 2, $p = 0.011$) and the distribution of the box ($\chi^2 = 8.429$, d.f. = 2, $p = 0.015$), respectively. However, the control procedure for maintaining the false discovery rate at 5% (see section 'Models' description') revealed that those aforementioned paired interactions are probably false positive, therefore they were discarded from the model and we considered independent variables for interpretability and limitation of overparametrization [52].

## 3.4. Change of the distribution

Our full model clearly differentiates from the null model, meaning that the previous choice (in terms of the distribution and whether the box was baited or not), together with species, rank and task/season impacted the decision whether the focal individual decided to leave the distribution or not (full/null model comparison: $\chi^2 = 220.040$, d.f. = 25, $p < 0.001$). Especially, whether the previous visited box was baited or not clearly impacted the probability to choose to leave the current distribution, with this probability being higher after choosing a non-baited box before (estimates $\pm$ s.d. = 1.453 $\pm$ 0.321, reference being baited box, $\chi^2 = 10.604$, d.f. = 1, $p = 0.001$; figure 3). The effects of the task/season ($\chi^2 = 7.797$, d.f. = 2, $p = 0.020$) and of the distribution (clumped versus scattered) which the previous box belonged to ($\chi^2 = 5.710$, d.f. = 1, $p = 0.017$) were identified as false positive according to Benjamini and Liu's false discovery rate correction approach. Note, however, that in this case, the probability to leave for the other food distribution decreased if the previous box belonged to the clumped distribution (estimates $\pm$ s.d. = $-0.740 \pm 0.441$, reference being the scattered distribution), and was the lowest in CvQ task (estimates $\pm$ s.d. = $-0.483 \pm 0.185$, reference being CvS1).

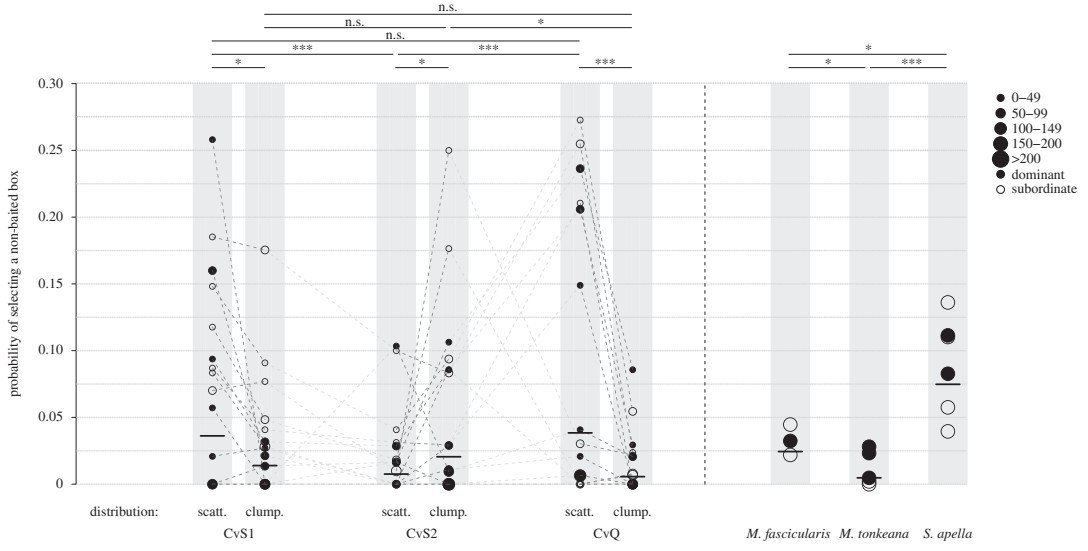

**Figure 4.** Influence of the location of the target box (on the left) and of the species (on the right) on the probability of selecting a non-baited box (model 'Goal-directed strategy'). scatt. = box selected belonged to scattered distribution, clump. = box selected belonged to clumped distribution. See legend of figure 2 for further information.

## 3.5. Goal-directed strategy

The model 'Goal-directed strategy' significantly differed from the corresponding null model (full/null model comparison: $\chi^2 = 126.550$, d.f. = 23, $p < 0.001$). The probability of choosing an empty box (thus that from another task/season) was influenced by the fruit distribution differentially across tasks/ seasons (task/distribution of box (clumped) interaction, $\chi^2 = 20.543$, d.f. = 2, $p < 0.001$, table 1; figure 4) and with differences among species ($\chi^2 = 21.469$, d.f. = 2, $p < 0.001$, figure 4). Specifically, in both CvS1 and CvQ (in which the two distributions where located in the same side of the outdoor area) primates visited more non-baited boxes when foraging in the scattered than in the clumped distribution (figure 4, left; table 1), while this was not the case for CvS2. In terms of species, *M. tonkeana* showed significantly more goal-directed movements while visiting baited boxes (with an 'error rate', i.e. visit of non-baited boxes, close to zero according to the model predictions) compared to *S. apella*. *Sapajus apella* visited non-baited boxes around five times more than *M. tonkeana* when following model predictions, while *M. fascicularis* significantly differed from the two aforementioned species with an intermediate error rate (figure 4, right; table 1). The learning variable ('session') showed that individuals had a lower tendency to visit non-baited boxes over time ($-0.361 \pm 0.087$, $\chi^2 = 9.137$, d.f. = 1, $p = 0.003$). Rank did not affect the overall visiting 'error rate' of baited boxes ($\chi^2 = 1.316$, d.f. = 1, $p = 0.251$). After visiting all six boxes of the clumped distribution (*M. tonkeana* = 44.0% of trials, *M. fascicularis* = 38.7% and *S. apella* = 31.6%), only *M. tonkeana* visited at least three baited boxes in the scattered distribution significantly more often than random: *M. tonkeana* in 83% of trials, ($N_{\text{trials}} = 206$, binomial test $p < 0.001$), *M. fascicularis* in 45.5% ($N_{\text{trials}} = 55$, $p = 0.600$) and *S. apella* in 58.2% ($N_{\text{trials}} = 91$, $p = 0.140$).

## 4. Discussion

In this study, we investigated spatial memory ability, particularly related to spatial food distribution, since each task/season lasted five days during which subjects were tested twice per day. All three primate species seem to have built a local spatial knowledge of the available fruit distributions which they could probably use for decision-making while foraging. Even though primates did not visit a box of the clumped distribution as the first feeding site significantly more often, results of the distribution preference (based on the Index of Lateralization) and of the sites visited after the first ('Successive five visited boxes') showed that all species (even if there was a less clear preference in *S. apella*) had a preference for feeding on fruit available in the clumped rather than in the scattered distribution in both conditions ('*Where*' and '*Where* versus *what*' Tasks). In general, this preference seemed to be stronger for the most frugivorous species, the Tonkean macaque, who also foraged with

more goal-directed movements (model 'Successive five boxes CvS' and model 'Goal-directed strategy'). Finally, we showed that individuals changed the current chosen distribution more often after choosing non-baited boxes ('Change of the distribution').

Food from a more densely arranged patch allows for the minimization of the energetic costs of animal foraging efforts [60]. In fact, a density bias, where densely arranged food items seem more numerous relative to the same amount of food sparsely arranged [61] was found in different primates (non-human primates [61,62]; humans [63,64]). This perception is important for higher survival and reproductive success [65,66]. In accordance with our results, captive and wild capuchin monkeys have already shown the ability to minimize travel costs while maximizing intake when feeding on fruit (e.g. [61,67,68]). On the contrary, previous studies on *M. mulatta* did not show preference for a densely arranged array compared to a sparsely arranged one in a virtual reality experiment [61]. This underlines again the importance of ecologically meaningful protocol designs to investigate animal foraging and planning behaviour.

All tested species generally prioritized the clumped distribution (*where*), thus maximizing foraging efficiency to the detriment of fruit preference or quality (*what*). However, the most frugivorous species, *M. tonkeana*, showed a stronger preference (compared to the other two species) for the clumped distribution (compared to the scattered one) when fruit quality did not differ in the two distributions (similarly to the results of the Index of Lateralization), but not when the most preferred fruit was in the scattered distribution. The erasure of such species differences in the 'Where versus *what*' task/season (CvQ) suggests that all species do not equally incorporate food preference information when making decisions while foraging. Therefore, even though maximization of foraging efficiency seems to be ubiquitously *a priori* factor affecting foraging decisions in primates, including provisioned primates (as also indicated by the similar intensity of the estimates in the models 'Successive five boxes CvS' and 'Successive five boxes CvQ'), fruit quality (in terms of preference) could still play a role in highly frugivorous primates [38]. Wild animals plan their travelling according to preferred resources and choose foods according to their respective nutritional value and their reciprocal availability in a given seasonal period (primates [69–73]; other mammals [74,75]). Formulating accurate judgements on food quantity and quality together is a crucial ability to forage efficiently [76].

Our findings support that more frugivorous species may rely more on local spatial memory of resources than less frugivorous species [17]. The less frugivorous species, *S. apella*, visited non-baited sites five times more than the two more frugivorous species who thus showed more goal-directed travel. *Sapajus apella* may thus possess less accurate spatial cognitive abilities for food location in comparison to the more frugivorous primates. However, this seems unlikely since they clearly used spatial memory information to forage efficiently by choosing the profitable clumped distribution, like the other species did. Thus, given their known ability to integrate the variables *what*, *where* and *when* in the wild [29], a second hypothesis could be that they possess a better large-scale memory compared to a small-scale one. However, the difference between the small and large scales in our protocol setting is not large in comparison to the average yearly home-range of this species (1.60 $\pm$ 77 km$^2$ [77]). These results and the recognized high cognitive abilities of *S. apella* among monkeys [78] render it unlikely that this species lacks small-scale spatial memory. The most likely explanation is that being more omnivorous/insectivorous than the other two study species (electronic supplementary material, table S1), they rely more on mobile resources (e.g. insects) thus revisiting potential feeding sites may be more profitable for them. Similarly, more frugivorous lemurs showed more goal-directed travels while more insectivorous lemurs showed a more exploratory behaviour [79]. Moreover, the probability that individuals adopted such an 'opportunistic behaviour' (i.e. visiting also other non-baited feeding sites) was higher for all three species (but only in two of the three tasks) when they fed in the scattered distribution. Animals are more likely to find new food and/or competitors [67] if they cover longer distances, making opportunistic feeding a more profitable strategy when food is scattered. In general, the results of the model 'Change of the distribution' show, however, that all three species had a higher probability to change distribution, thus re-correct themselves, after choosing a non-baited box. This result suggests that primates readjust their strategy based on trial-error processes. This behaviour may help to explain why we did not find significant results for the model 'First visited box' but we found a general preference for the clumped distribution using the Index of Lateralization. Indeed, primates may still memorize the working scenarios with the underlying idea of two existing fruit patches in which foraging costs differ: one with potentially more distant boxes, and therefore error-prone, or another with closer boxes. If so, making a mistake could be an indication of being in the more error-prone patch and that signal would be processed, resulting in a change of foraging distribution. However, this is logical only if primates could at least empirically

determine differences among patches in the local probability of finding a non-baited box. Therefore, it leaves open the questions of the cognitive mechanisms that are at play in primates' foraging decision-making, especially with regards to orientation/re-orientation events. Furthermore, other variables that we could not control for in our semi-free ranging protocol may have affected initial foraging choices of the individuals. First, first visited boxes may have been affected by the presence of competitors at the boxes of the non-chosen distribution. Overall, we did not find any effect of social competition on the foraging decisions even for the more despotic species (*M. fascicularis*; electronic supplementary material, table S1). However, in our experimental protocol we could test only for the influence of the number of individuals that were present around the *chosen* box since we could not collect information on the presence of (dominant) individuals at the *non-chosen* boxes, thus at the non-chosen fruit distribution. Therefore, assessing solely the first feeding choice in our study design where we could only *partially* test and control for social competition, it may not allow depicting true feeding decisions.

While considering these issues, our results from the Index of Lateralization and the GLMMs suggest that primates remembered the position of the baited boxes and of the most profitable distribution. It can be argued that this general preference towards the clumped distribution could have been due to our experimental settings that provided a local difference in baited box density once the subject chose to forage on a given distribution: i.e. higher density of baited boxes within the clumped distribution in comparison to the scattered one (see map of the outdoor experiments, figure 1*a*–*c*). If such, we would have expected strong differences in the tendency to change distribution, with this probability being much higher for changes from the scattered to the clumped one than the reverse. Our results show that, if individuals chose a non-baited box, they then changed the distribution they were foraging in more than when opening a baited box (model 'Change of the distribution'). It is important to note that the distribution of the previous box had no clearly significant effect in interaction with the nature of the box (i.e. baited or not), nor independently (both were identified as highly likely false positive). These results therefore help in interpreting the overall preference for the clumped distribution assessed with the Index of Lateralization, since it helps excluding the hypothesis of a local density bias of boxes as the distribution has weak to no effect. However, it leaves still open the questions of the cognitive mechanisms that are involved in primate foraging decision-making, especially with regard to orientation/re-orientation events.

Finally, after feeding from the clumped fruit, only the most frugivorous species fed at above chance level also from the scattered fruit (*M. tonkeana* = 83%), rather than continuing feeding within/near the clumped distribution (i.e. visiting empty boxes from other tasks/seasons). Since the notions of *where* (space) and *when* (time) in humans, share a common representational format in the spatial configuration and are strongly interconnected in the mind circuits [80], this result may suggest that more frugivorous primates, or at least Tonkean macaques, possess spatio-temporal knowledge of food location. This result may also be explained by the combination of different factors such as higher motivation for more frugivorous primates to pursue fruit foraging, higher competition in more despotic societies (for which we could not fully control, as mentioned above), and a less goal-directed foraging strategy in less frugivorous species.

Our study highlighted that species with different degrees of frugivory possess the ability to discriminate and remember the position of more and less profitable available feeding sites. However, given the small sample size in number of individuals tested per species, any generalization of the results at the species level are tentative, and further studies are needed to confirm the observed trend. The ability to discriminate the relative quantity (density bias) is a crucial cognitive ability that is also present in distantly related species to primates [81,82]. Taken together, 'Where memory' and 'Density bias' are crucial cognitive skills that influence the foraging decisions of frugivorous primates to maximize energy intake and decrease foraging effort [5]. It is worth mentioning that animals in our study had to choose between palatable and attractive fruits with fairly comparable nutritional values (electronic supplementary material, table S8). With a greater variation in food quality, *what* would certainly play a more important role in primate foraging decisions. As a growing body of research on primates [17,20,30–32] and other animals (fish [83]; birds [84]; carnivores [85]) indicates, our study underlines how the feeding ecology of a species affects the evolution of its abstract mental abilities (*Ecological Intelligence Hypothesis*, e.g. [13,20]) and, more generally, its behaviour [86,87]. Particularly, this underlines how a more frugivorous diet can be linked to higher cognitive abilities, memory capacities and even hippocampal size [83,88]. Increased foraging efficiency in early humans has been suggested to be an important evolutionary step enabling encephalization during human evolution [89]. Larger cross-species comparisons could help us building a picture of the functional causes of the evolution of the different foraging strategies adopted by animals [90].

**Ethics.** This study was performed according to the French legal requirements for the use of animals in research and to the EU Directive 2010/63/EU on the welfare of animals used for scientific purposes. Subject participation was voluntary, and they continued with their normal feeding routine during testing. Subjects had ad libitum access to water and were never food or water deprived during this study.

**Data accessibility.** The dataset supporting this article is available from the Dryad Digital Repository at: https://doi.org/10.5061/dryad.fk334rr [44].

**Authors' contributions.** S.M. and H.M. conceived the study; C.T. participated in designing it. C.T., G.T. and S.d.G. collected the data. B.R. and C.T. analysed the data with the help of S.M. C.T., B.R. and S.M. wrote the manuscript, H.M. edited it. G.T. helped with formatting. All authors gave final approval for publication.

**Competing interests.** The authors declare no competing interests.

**Funding.** This work was supported by Fondazione Ethoikos (Radicondoli, SI, Italy) who supported the PhD stipend for Cinzia Trapanese, the Primate Center of Strasbourg University, the Société Francophone De Primatologie (SFDP) and the UMR 7206 Eco-anthropologie (Muséum national d'Histoire naturelle-CNRS-Univ. Paris 7). B.R. was supported by a studentship grant from the École Normale Supérieure de Paris.

**Acknowledgements.** We are sincerely grateful to R. Cozzolino (Ethoikos Foundation), to the UMR 7206 Eco-anthropologie (Muséum national d'Histoire naturelle-CNRS-Univ. Paris 7), to the Société Francophone De Primatologie (SFDP) and to the Primate Center of Strasbourg University for financial support. Thank you to Y. Larmet and his team for allowing us to conduct the study at the Primate Center of Strasbourg University. We also thank J. Wittemer for logistical aid, all caretakers and veterinarians for their helpful assistance throughout the study, the numerous trainees that helped with the data collection (A. Sotto-Mayor, P. Ibáñez de Aldecoa, L. Berrod, L. Marmol-Gomez, P. Meunier, M.L. Poiret, Y. Chantrel, C. Lepas, M. Bey, A. Navarra, M. Hirel, T. Leclere and I. Roho). We also thank the European Erasmus Program and the Muséum national d'Histoire naturelle for supporting some of them. Thanks to C. de Lillo and J.C. Cassel for their advice and to L. Muniz for her feedback on the manuscript and the English editing.

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
