## [Reviewer comments · Royal Society Open Science]

Review History

RSOS-181722.R0 (Original submission)

Review form: Reviewer 1 (Yumiko Yamazaki)

Is the manuscript scientifically sound in its present form?

Yes

Are the interpretations and conclusions justified by the results?

No

Is the language acceptable?

Yes

Is it clear how to access all supporting data?

Yes

Do you have any ethical concerns with this paper?

No

Have you any concerns about statistical analyses in this paper?

I do not feel qualified to assess the statistics

Recommendation?

Accept with minor revision (please list in comments)

Comments to the Author(s)

Comments can be found in an attached file (Appendix A).

Review form: Reviewer 2

Is the manuscript scientifically sound in its present form?

No

Are the interpretations and conclusions justified by the results?

No

Is the language acceptable?

No

Is it clear how to access all supporting data?

Yes

Do you have any ethical concerns with this paper?

No

Have you any concerns about statistical analyses in this paper?

Yes

Recommendation?

Major revision is needed (please make suggestions in comments)

Comments to the Author(s)

In this paper, the authors compare the foraging decisions of three species of captive primates, two macaques and one capuchin species, on two food arrays, one clumped and one dispersed, baited with foods of different preference. They use this set up to mimic foraging in a seasonal environment and they also looked at the degree that travel was goal-directed within the experiment. They interpret their results as showing that all three species preferred the clumped array. However, this preference was not as strong for the most frugivorous species when preferred fruit was scattered. This species also used more goal-directed travel. The authors interpret this to show that the most frugivorous species was the most efficient foraging taking into consideration what food was where when making decisions.

There seems to be valuable data here but the description of the experiments and the presentation of the results are often confusing and it is difficult to understand exactly what has been done. I

also have some serious issues with the interpretation of the Results that I will detail below. I am not sure that the authors actually found what they say they found.

As a first point, I really think that the title needs revision. Most people will not understand right away what is meant by “miniature nature”. Are these animals semi-free ranging? Or should they be considered captive? I would leave this out of the title. The word “triggers” is also contentious in the title. It implies causation but this is not shown by the data. A better title would be something like, “Where and What? Greater frugivory is associated with more efficient foraging in three semi-free ranging monkey species”.

Early in the Introduction of the species tested, some information should be included on the social organization and dominance regimes of these species. Later in the discussion some conclusions are presented based on which species is more despotic and which is more tolerant but these features are not introduced.

At the end of the Introduction, it would be helpful if what was done was clarified more. For instance, on line 81, a “miniature nature protocol” should be defined. Later, on line 84 and below, it needs to be stated whether animals were tested alone or in groups. If in groups, how was the competition with other group members accounted for? In addition, the reasoning behind the prediction made on lines 86-89 should be presented. Why was the more omnivorous species not expected to feed on the scattered, preferred food first?

In the Methods, the description of the models that were run is very confusing. Especially where interaction terms and comparisons of models up. I think this could be rectified by amalgamating all the information provided in the Description of Statistical Models section (L 230-253) into the introduction of each model earlier. If each model were discussed fully, in turn, it may help the reader significantly.

Another point that would improve the readers understanding of what was done would be to provide Figures S6a and b and S7 within the main manuscript rather than in the ESM. These are vital to understanding the set up and the reader should not have to search for them.

My main issue with the interpretation of the Results is that I am not convinced that the monkeys actually preferred the clumped distribution, which is a key finding from the study. The first visited box was not in the clumped distribution significantly more often. Even though the five successively visited boxes were more likely to be from the clumped distribution if the first choice was from that distribution, this does not tell us anything. Once the animals had made their initial decision, it is not surprising that they then chose food sites that were nearby. There is a large set of literature showing distance minimizing in primates and other animals while foraging. In Fig. S6a and b and S7, it seems that sites within each distribution are closer to one another than the distance of the distributions from one another (exact distances should be provided). A model could be run to show whether, when choosing the scattered distribution first, the monkeys then stayed within that distribution for the next successive five choices. That could perhaps tell you if they foraged to decrease the distance traveled between sites (which is still somewhat interesting), even if they did not make the best choice initially. The authors may have to change the focus of the study slightly or find another way to convince the reader that the clumped distribution was actually chosen more often.

Minor comments:

L 20-21 – It would sound better if the first two sentences were linked by adding an “and” after “challenging”. Also, add “have” after “studies” and change “to” to “with”.

L 24 – Was the *Sapajus* sp. not known or what this a mix of animals from different species? It

would good if this could be clarified later in the Methods.

L 28 - Change "Primates" to "All species", otherwise it is not clear whether species or individuals are being discussed.

L 28 - Change "with" to "but".

L 29 - Add "did so" before "at".

L 30 - Delete "using spatial memory" here since this is a supposition.

L 33 - Change "the environment" to be more specific. Maybe to something like "food type and distribution".

L 34 - It is unclear how these results impact conservation. Please be more explicit or delete this.

L 38 - Change "arduousness" to "difficulties involved".

L 39 - Change "might be due to the" to "vary depending on qualities like a".

L 40 - Change "the" to "a".

L 41-42 - These paragraphs should be joined.

L 42-43 - This sentence is confusing. Please rewrite as, "To maximize foraging efficiency, animals should attempt to increase energetic intake and minimize costs".

L 44 - Change "tropical forests" to "seasonal environments". After all, truly tropical forests are notable for their lack of seasonality.

L 44 - "this dense environment" to "forests".

L 45 - Delete "seasonal and" since it is redundant.

L 48 - Change "characterized by a" to "mean there is".

L 49 - Change the first "and" to "with".

L 52 - Start this sentence with "Evolution of the ability".

L 57 - Add "have" after "mammals".

L 28-29 - Move the second "fruit" to after "patchy" and change "phenology" to "availability".

L 60 - Change "constantly" to "consistently" and add "and" before "hence".

L 61 - Delete "noticeable".

L 65 - For clarity, consider changing the middle of this sentence to, "in shaping the evolution of cognitive complexity has gained precedence over the view...".

L 73 - Add "to" after "planning".

L 74 - Change "revisit" to "revisiting".

L 77 - Change the middle of the sentence to, "allow the manipulation of variables...".

L 78 - Add "the" after "compared" and "strategies" after "foraging".

L 79 - Change "different" to "differing".

L 81 - Please define what a "miniature nature" protocol is.

L 83 - Change "this" to "the".

L 84 - Change "is" to "was", change "and" to "versus", and change "predict" to "predicted".

L 85 - Add "would" after "primates".

L 86 - Change "is" to "was". These types of predictions are usually best in past tense throughout, since the work is already done.

L 87 - Change "is" to "was" and change "predict" to "predicted".

L 88 - Add "would" before "feed" and "resource" after "this".

L 90 - Change "are" to "were".

L 91 - What does "feeding sites" here refer to? Each food box? Or the overall array?

L 102 - Change "spontaneous" to "natural".

L 103 - Move "daily" to after "fed", pluralized "pellet", and change "provision" to "provisioning".

L 104 - The part here "other fruit types have been used in experiments" is not clear. Do you mean that you did not want to over-feed them or was this to encourage participation in the experiments? Or both? Please clarify.

L 106 - Delete "to be" and change the second "the" to "these".

L 107 - Delete "to" and add "with" after "individuals".

L 112 - Add "a" after "from".

L 114 - Since all boxes remained in the outdoor areas even when they were not baited, please

clarify here how the monkeys were to know what boxes were baited. Was this based on memory of a previous trial? This needs to be explicit.

L 119 – What does “(two)” refer to? To fruits or two sites?

L 119 – Also please be explicit here as to why you pretended to fill 12 boxes. What were the monkeys supposed to get from this?

L 143 – So the experiment took 16-weeks total. This information should be included here.

L 145 – Delete “the primate”.

L 149 – It says here the monkeys were “closely followed”. Please state how close and how you made sure that this did not influence the decisions and behavior of the animals.

L 154 – What does the “red line” refer to? Since you haven’t defined it yet, please put “(see below)” or something equivalent here.

L 156 – Add “to” after “way”.

L 158 – Change “this” to “competition by”.

L 172-173 – This sentence should be broken up and rearranged to be clearer. “In CvS2, the relative positions of the clumped and scattered distributions was inverted.” Then describe the fruit that was available where. This part is very confusing. Does “similarly preferred” mean the same? And what do you mean by “only available in other boxes”?

L 184 – Please provide a formula for the Index of lateralization.

L 186 – Change “area” to “enclosure”.

L 194 – Change “consisted in” to “was”.

L 206 – Start a new paragraph here.

L 208 – Does similar spatial configuration refer to the same one? Not clear.

L 212-215 – The description of the models is so confusing.

L 217 – I still don’t understand how the monkeys were supposed to know which boxes were unbaited. I see that this is stated in the ESM but a brief statement at the end of the Introduction in the paper is also needed.

L 243 – Add “or below” after “at”.

L 268 – Move “first” to after “distribution”.

L 277 – Change “and” to “or”.

L 278 – Add “were found either” after the parentheses.

L 280 – Please remind the reader here that the scattered distribution had more preferred food.

L 293 – Begin a new paragraph here.

L 300 – Please add more information as to how competitors affected foraging decisions.

L 302 – Please explain how long-tailed macaques were more impacted than the other two species.

L 312 – Delete “high”.

L 313 – Change “closed” to “close”.

L 315 – Pluralize “prediction”.

L 317 – Add “a” after “had”.

L 320 – Change “removed” to “remove”.

L 327 – Change the comma to a period.

L 330-338 – I don’t think that it has been shown that the clumped distribution was actually chosen more.

L 342 – Add a comma after “bias”.

L 344 – “crucial” seems like an overstatement.

L 348 – Add an “a” after “in”.

L 351 – Change “maximize” to “maximizing”.

L 352 – Change “in” to “to the”.

L 355 – “a priori” should be “a priori” in italics.

L 363 – Change “less” to “least”.

L 365 – Add “a” after “possess”.

L 369 – Add “better” before “large”.

L 370 – Change “but not a” to “compared to a”.

L 373 – Change the end of the sentence to, “render it unlikely that this species lacks small-scale

spatial memory”.

L 381 – Delete the third “the”.

L 384 – Delete “also”.

L 385-386 – This sentence on humans should be deleted. It doesn’t add anything and seems tacked on.

L 390-391 – This sentence needs more explanation.

L 392 – Change “Tonkeana” to “Tonkean macaques”.

L 394 – Change “strategy” to “strategies”.

L 395 – Start a new paragraph here.

L 398 – Change “from” to “is also present in”.

L 399 – Add “the” after “influence”.

L 400 – Change “to mention” to “mentioning”.

L 411-414 – I don’t really see how this relates to conservation strategies. This line is simplistic and should probably be deleted.

Decision letter (RSOS-181722.R0)

23-Jan-2019

Dear Dr Trapanese,

The editors assigned to your paper ("Where or What? Primates in “Miniature Nature”: frugivory triggers spatial cognition to forage efficiently") have now received comments from reviewers. We would like you to revise your paper in accordance with the referee and Associate Editor suggestions which can be found below (not including confidential reports to the Editor). Please note this decision does not guarantee eventual acceptance.

Please submit a copy of your revised paper before 15-Feb-2019. Please note that the revision deadline will expire at 00.00am on this date. If we do not hear from you within this time then it will be assumed that the paper has been withdrawn. In exceptional circumstances, extensions may be possible if agreed with the Editorial Office in advance. We do not allow multiple rounds of revision so we urge you to make every effort to fully address all of the comments at this stage. If deemed necessary by the Editors, your manuscript will be sent back to one or more of the original reviewers for assessment. If the original reviewers are not available, we may invite new reviewers.

- Data accessibility

<http://datadryad.org/submit?journalID=RSOS&manu=RSOS-181722>

- Competing interests

- Authors' contributions

- Acknowledgements

- Funding statement

Once again, thank you for submitting your manuscript to Royal Society Open Science and I look

forward to receiving your revision. If you have any questions at all, please do not hesitate to get in touch.

on behalf of Dr Atsushi Iriki (Associate Editor) and Professor Kevin Padian (Subject Editor)
 openscience@royalsociety.org

Comments to Author:

Reviewers' Comments to Author:

Reviewer: 1

Comments to the Author(s)

Comments can be found in an attached file

Reviewer: 2

Comments to the Author(s)

In this paper, the authors compare the foraging decisions of three species of captive primates, two macaques and one capuchin species, on two food arrays, one clumped and one dispersed, baited with foods of different preference. They use this set up to mimic foraging in a seasonal environment and they also looked at the degree that travel was goal-directed within the experiment. They interpret their results as showing that all three species preferred the clumped array. However, this preference was not as strong for the most frugivorous species when preferred fruit was scattered. This species also used more goal-directed travel. The authors interpret this to show that the most frugivorous species was the most efficient foraging taking into consideration what food was where when making decisions.

There seems to be valuable data here but the description of the experiments and the presentation of the results are often confusing and it is difficult to understand exactly what has been done. I also have some serious issues with the interpretation of the Results that I will detail below. I am not sure that the authors actually found what they say they found.

As a first point, I really think that the title needs revision. Most people will not understand right away what is meant by "miniature nature". Are these animals semi-free ranging? Or should they be considered captive? I would leave this out of the title. The word "triggers" is also contentious in the title. It implies causation but this is not shown by the data. A better title would be something like, "Where and What? Greater frugivory is associated with more efficient foraging in three semi-free ranging monkey species".

Early in the Introduction of the species tested, some information should be included on the social organization and dominance regimes of these species. Later in the discussion some conclusions are presented based on which species is more despotic and which is more tolerant but these features are not introduced.

At the end of the Introduction, it would be helpful if what was done was clarified more. For instance, on line 81, a "miniature nature protocol" should be defined. Later, on line 84 and below, it needs to be stated whether animals were tested alone or in groups. If in groups, how was the

competition with other group members accounted for? In addition, the reasoning behind the prediction made on lines 86-89 should be presented. Why was the more omnivorous species not expected to feed on the scattered, preferred food first?

In the Methods, the description of the models that were run is very confusing. Especially where interaction terms and comparisons of models up. I think this could be rectified by amalgamating all the information provided in the Description of Statistical Models section (L 230-253) into the introduction of each model earlier. If each model were discussed fully, in turn, it may help the reader significantly.

Another point that would improve the readers understanding of what was done would be to provide Figures S6a and b and S7 within the main manuscript rather than in the ESM. These are vital to understanding the set up and the reader should not have to search for them.

My main issue with the interpretation of the Results is that I am not convinced that the monkeys actually preferred the clumped distribution, which is a key finding from the study. The first visited box was not in the clumped distribution significantly more often. Even though the five successively visited boxes were more likely to be from the clumped distribution if the first choice was from that distribution, this does not tell us anything. Once the animals had made their initial decision, it is not surprising that they then chose food sites that were nearby. There is a large set of literature showing distance minimizing in primates and other animals while foraging. In Fig. S6a and b and S7, it seems that sites within each distribution are closer to one another than the distance of the distributions from one another (exact distances should be provided). A model could be run to show whether, when choosing the scattered distribution first, the monkeys then stayed within that distribution for the next successive five choices. That could perhaps tell you if they foraged to decrease the distance traveled between sites (which is still somewhat interesting), even if they did not make the best choice initially. The authors may have to change the focus of the study slightly or find another way to convince the reader that the clumped distribution was actually chosen more often.

Minor comments:

L 20-21 - It would sound better if the first two sentences were linked by adding an "and" after "challenging". Also, add "have" after "studies" and change "to" to "with".

L 24 - Was the Sapajus sp. not known or what this a mix of animals from different species? It would good if this could be clarified later in the Methods.

L 28 - Change "Primates" to "All species", otherwise it is not clear whether species or individuals are being discussed.

L 28 - Change "with" to "but".

L 29 - Add "did so" before "at".

L 30 - Delete "using spatial memory" here since this is a supposition.

L 33 - Change "the environment" to be more specific. Maybe to something like "food type and distribution".

L 34 - It is unclear how these results impact conservation. Please be more explicit or delete this.

L 38 - Change "arduousness" to "difficulties involved".

L 39 - Change "might be due to the" to "vary depending on qualities like a".

L 40 - Change "the" to "a".

L 41-42 - These paragraphs should be joined.

L 42-43 - This sentence is confusing. Please rewrite as, "To maximize foraging efficiency, animals should attempt to increase energetic intake and minimize costs".

L 44 - Change "tropical forests" to "seasonal environments". After all, truly tropical forests are notable for their lack of seasonality.

L 44 - "this dense environment" to "forests".

- L 45 - Delete "seasonal and" since it is redundant.
- L 48 - Change "characterized by a" to "mean there is".
- L 49 - Change the first "and" to "with".
- L 52 - Start this sentence with "Evolution of the ability".
- L 57 - Add "have" after "mammals".
- L 28-29 - Move the second "fruit" to after "patchy" and change "phenology" to "availability".
- L 60 - Change "constantly" to "consistently" and add "and" before "hence".
- L 61 - Delete "noticeable".
- L 65 - For clarity, consider changing the middle of this sentence to, "in shaping the evolution of cognitive complexity has gained precedence over the view...".
- L 73 - Add "to" after "planning".
- L 74 - Change "revisit" to "revisiting".
- L 77 - Change the middle of the sentence to, "allow the manipulation of variables...".
- L 78 - Add "the" after "compared" and "strategies" after "foraging".
- L 79 - Change "different" to "differing".
- L 81 - Please define what a "miniature nature" protocol is.
- L 83 - Change "this" to "the".
- L 84 - Change "is" to "was", change "and" to "versus", and change "predict" to "predicted".
- L 85 - Add "would" after "primates".
- L 86 - Change "is" to "was". These types of predictions are usually best in past tense throughout, since the work is already done.
- L 87 - Change "is" to "was" and change "predict" to "predicted".
- L 88 - Add "would" before "feed" and "resource" after "this".
- L 90 - Change "are" to "were".
- L 91 - What does "feeding sites" here refer to? Each food box? Or the overall array?
- L 102 - Change "spontaneous" to "natural".
- L 103 - Move "daily" to after "fed", pluralized "pellet", and change "provision" to "provisioning".
- L 104 - The part here "other fruit types have been used in experiments" is not clear. Do you mean that you did not want to over-feed them or was this to encourage participation in the experiments? Or both? Please clarify.
- L 106 - Delete "to be" and change the second "the" to "these".
- L 107 - Delete "to" and add "with" after "individuals".
- L 112 - Add "a" after "from".
- L 114 - Since all boxes remained in the outdoor areas even when they were not baited, please clarify here how the monkeys were to know what boxes were baited. Was this based on memory of a previous trial? This needs to be explicit.
- L 119 - What does "(two)" refer to? To fruits or two sites?
- L 119 - Also please be explicit here as to why you pretended to fill 12 boxes. What were the monkeys supposed to get from this?
- L 143 - So the experiment took 16-weeks total. This information should be included here.
- L 145 - Delete "the primate".
- L 149 - It says here the monkeys were "closely followed". Please state how close and how you made sure that this did not influence the decisions and behavior of the animals.
- L 154 - What does the "red line" refer to? Since you haven't defined it yet, please put "(see below)" or something equivalent here.
- L 156 - Add "to" after "way".
- L 158 - Change "this" to "competition by".
- L 172-173 - This sentence should be broken up and rearranged to be clearer. "In CvS2, the relative positions of the clumped and scattered distributions was inverted." Then describe the fruit that was available where. This part is very confusing. Does "similarly preferred" mean the same? And what do you mean by "only available in other boxes"?
- L 184 - Please provide a formula for the Index of lateralization.

- L 186 - Change "area" to "enclosure".
- L 194 - Change "consisted in" to "was".
- L 206 - Start a new paragraph here.
- L 208 - Does similar spatial configuration refer to the same one? Not clear.
- L 212-215 - The description of the models is so confusing.
- L 217 - I still don't understand how the monkeys were supposed to know which boxes were unbaited. I see that this is stated in the ESM but a brief statement at the end of the Introduction in the paper is also needed.
- L 243 - Add "or below" after "at".
- L 268 - Move "first" to after "distribution".
- L 277 - Change "and" to "or".
- L 278 - Add "were found either" after the parentheses.
- L 280 - Please remind the reader here that the scattered distribution had more preferred food.
- L 293 - Begin a new paragraph here.
- L 300 - Please add more information as to how competitors affected foraging decisions.
- L 302 - Please explain how long-tailed macaques were more impacted than the other two species.
- L 312 - Delete "high".
- L 313 - Change "closed" to "close".
- L 315 - Pluralize "prediction".
- L 317 - Add "a" after "had".
- L 320 - Change "removed" to "remove".
- L 327 - Change the comma to a period.
- L 330-338 - I don't think that it has been shown that the clumped distribution was actually chosen more.
- L 342 - Add a comma after "bias".
- L 344 - "crucial" seems like an overstatement.
- L 348 - Add an "a" after "in".
- L 351 - Change "maximize" to "maximizing".
- L 352 - Change "in" to "to the".
- L 355 - "a prior" should be "a priori" in italics.
- L 363 - Change "less" to "least".
- L 365 - Add "a" after "possess".
- L 369 - Add "better" before "large".
- L 370 - Change "but not a" to "compared to a".
- L 373 - Change the end of the sentence to, "render it unlikely that this species lacks small-scale spatial memory".
- L 381 - Delete the third "the".
- L 384 - Delete "also".
- L 385-386 - This sentence on humans should be deleted. It doesn't add anything and seems tacked on.
- L 390-391 - This sentence needs more explanation.
- L 392 - Change "Tonkeana" to "Tonkean macaques".
- L 394 - Change "strategy" to "strategies".
- L 395 - Start a new paragraph here.
- L 398 - Change "from" to "is also present in".
- L 399 - Add "the" after "influence".
- L 400 - Change "to mention" to "mentioning".
- L 411-414 - I don't really see how this relates to conservation strategies. This line is simplistic and should probably be deleted.

Author's Response to Decision Letter for (RSOS-181722.R0)

See Appendix B.

RSOS-181722.R1 (Revision)

Review form: Reviewer 2

Is the manuscript scientifically sound in its present form?

Yes

Are the interpretations and conclusions justified by the results?

Yes

Is the language acceptable?

Yes

Is it clear how to access all supporting data?

Yes

Do you have any ethical concerns with this paper?

No

Have you any concerns about statistical analyses in this paper?

No

Recommendation?

Accept with minor revision (please list in comments)

Comments to the Author(s)

The authors have done an excellent job of addressing most of the issues I had with the first draft. The manuscript, and what exactly was done, is now much clearer. Given the new models they included, I am now convinced that the monkeys preferred to forage in the clumped distribution, even though they did not choose it initially significantly more often. A bit of work is still needed to better define the predictions (see below) and clean up the English but after that it should be a nice addition to the literature on decision-making during foraging.

L 49 - Delete "a", since the predictability is more likely annual rather than once in a lifetime.

L 72 - Reverse "feeding" and "past" to read "past feeding".

L 91 - I do not think "scattered" should be used here to refer to fruit trees given how it was used in the previous sentence and in the rest of the predictions. Perhaps this is better "Since fruit trees provide ephemeral foods located in variable distributions,".

L 94 - Please define what you mean by "more affected". Exactly how was their behavior expected to change?

L 95 - Change "do" to "did".

L 104 - Add "The" before "Sapajus" and change "like" to "likely".

L 107 - Delete "N=14" here because this not your sample size per species.

L 107 - Change "fully" to "full".

- L 114 – Change “primate” to “subject”.
- L 114-115 – Change “too large intakes” to “excessive intake”.
- L 119 – The beginning of this sentence is poorly-written. Change to “Since the three tested species are mainly frugivorous...”.
- L 128 – Add a comma after “test”.
- L 151 – Delete the “to”.
- L 159 – Change “that were not any more available” to “, which were no long available”.
- L 163 – Change “to” to “in”.
- L 173 – Change “on” to “as to”.
- L 179 – Delete “on”.
- L 192 – Change “because of” to “due to”.
- L 193 – Change “have been” to “were”.
- L 205 – Change “the each other position” to “one another”.
- L 256 – Change “fist” to “first”.
- L 258 – Change “have” to “had”.
- L 269 – Change “Since we” to “but”.
- L 277 – Change “tend” to “tended”.
- L 278 – Change “forage on” to “were foraging in”.
- L 280 – Change “previous” to “previously”.
- L 284 – Change “fine” to “fine-scale”.
- L 286 – Delete “than”.
- L 300-301 – Change “despotic degree” to “degree of despotism” and “potentially ultimately affecting” to “which could potentially affect their”.
- L 325 – Add “that” after “hypothesized”.
- L 326 – Change “being” to “may be”.
- L 328 – Change “if the variable species were as” to “whether each species was equally”.
- L 329 – Change “Like so” to “Thus”.
- L 338 – Change “were the individuals” to “they were”.
- L 340 – Change “had not” to “did not have the”.
- L 342 – The “it” here, or what is being tested, needs to be repeated for clarity.
- L 350 – The reference needs a closing parentheses.
- L 351 – I think “necessaries” needs to be changed to “necessary” here.
- L 353 – Change “variables” to “variable”.
- L 361-362 – Change the end of this sentence to “sequentially removing non-significant terms of higher complexity”.
- L 365 – Add “a” after “we”.
- L 371- Pluralize “effect”.
- L 375 – Change again to “sequentially removing non-significant terms of higher complexity”.
- L 379 – Here there is a double-negative. Do you mean “0 for no competitor present and 1 if at least one competitor was present”?
- L 380 – Pluralize “integer”.
- L 384 – Deleted “deeply”.
- L 385 – Change the middle of this sentence to, “did not seem to suffer from the reduced sample”.
- L 388 – Delete “the needed”.
- L 396 and 397 – I’m not sure I understand what “handed on” means in this context.
- L 408 – Change “of” to “from”.
- L 455 – Change “to notice” to “noticing”.
- L 471 – Change “when having faced” to “after choosing”.
- L 491 – The way this is written with the colon and the “they” reads as if *M. tonkeana* visited non-baited boxes 5-times more, which can’t be right...?
- L 504 – This would read better if a comma was put after “study” and “the” was deleted.
- L 508-509 – Delete “significantly more” and put “significantly more often” after “site”.
- L 510 – Change “with” to “there was”.

L 512 – Add “rather” after “clumped”.

L 514 and 558-561 – These results bring to mind a new paper on foraging site selection in lemurs which also found that more frugivorous species were more goal-directed and more insectivorous species are more exploratory, which is in line with your findings (Teichroeb & Vining, 2019, Navigation strategies in three nocturnal lemur species: Diet predicts heuristic use and degree of exploratory behavior. *Animal Cognition*).

L 516 – Change “when” to “after”.

L 523 – Change “showed” to “shown” and “travelling cost” to “travel costs”.

L 528 – Add “and” before “planning”.

L 562 – Please define what you mean by “opportunistic behavior” here.

L 566 – Change “have” to “had a”.

L 567-568 – Change “may pinpoint” to “suggest”.

L 599 – Changed “faced” to “chose” and “change more” to “changed the distribution they were foraging in more”.

L 603 – Add “in” after “help”.

L 608 – Add “at” after “fed”.

L 616 – Change “pursuit the” to “pursue”.

L 617 – Add “a” after “and”.

Decision letter (RSOS-181722.R1)

28-Mar-2019

Dear Dr Trapanese:

On behalf of the Editors, I am pleased to inform you that your Manuscript RSOS-181722.R1 entitled "Where and What? Frugivory is associated with more efficient foraging in three semi-free ranging primate species" has been accepted for publication in Royal Society Open Science subject to minor revision in accordance with the referee suggestions. Please find the referees' comments at the end of this email.

The reviewers and Subject Editor have recommended publication, but also suggest some minor revisions to your manuscript. Therefore, I invite you to respond to the comments and revise your manuscript.

- Ethics statement

- Data accessibility

It is a condition of publication that all supporting data are made available either as supplementary information or preferably in a suitable permanent repository. The data accessibility section should state where the article's supporting data can be accessed. This section should also include details, where possible of where to access other relevant research materials such as statistical tools, protocols, software etc can be accessed. If the data has been deposited in an external repository this section should list the database, accession number and link to the DOI for all data from the article that has been made publicly available. Data sets that have been

deposited in an external repository and have a DOI should also be appropriately cited in the manuscript and included in the reference list.

If you wish to submit your supporting data or code to Dryad (<http://datadryad.org/>), or modify your current submission to dryad, please use the following link:
<http://datadryad.org/submit?journalID=RSOS&manu=RSOS-181722.R1>

- **Competing interests**

- **Authors' contributions**

- **Acknowledgements**

- **Funding statement**

Because the schedule for publication is very tight, it is a condition of publication that you submit the revised version of your manuscript before 06-Apr-2019. Please note that the revision deadline will expire at 00.00am on this date. If you do not think you will be able to meet this date please let me know immediately.

When submitting your revised manuscript, you will be able to respond to the comments made by the referees and upload a file "Response to Referees" in "Section 6 - File Upload". You can use this

to document any changes you make to the original manuscript. In order to expedite the processing of the revised manuscript, please be as specific as possible in your response to the referees.

on behalf of Dr Atsushi Iriki (Associate Editor) and Professor Kevin Padian (Subject Editor)
openscience@royalsociety.org

Associate Editor Comments to Author (Dr Atsushi Iriki):

The authors have well addressed Reviewer 1's comments for minor revision, and most of Reviewer 2's major comments leaving a few minor clarification points. This manuscript seems ready for publication once gone through these points.

Reviewer comments to Author:

Reviewer: 2

Comments to the Author(s)

The authors have done an excellent job of addressing most of the issues I had with the first draft. The manuscript, and what exactly was done, is now much clearer. Given the new models they

included, I am now convinced that the monkeys preferred to forage in the clumped distribution, even though they did not choose it initially significantly more often. A bit of work is still needed to better define the predictions (see below) and clean up the English but after that it should be a nice addition to the literature on decision-making during foraging.

L 49 - Delete "a", since the predictability is more likely annual rather than once in a lifetime.

L 72 - Reverse "feeding" and "past" to read "past feeding".

L 91 - I do not think "scattered" should be used here to refer to fruit trees given how it was used in the previous sentence and in the rest of the predictions. Perhaps this is better "Since fruit trees provide ephemeral foods located in variable distributions,".

L 94 - Please define what you mean by "more affected". Exactly how was their behavior expected to change?

L 95 - Change "do" to "did".

L 104 - Add "The" before "Sapajus" and change "like" to "likely".

L 107 - Delete "N=14" here because this not your sample size per species.

L 107 - Change "fully" to "full".

L 114 - Change "primate" to "subject".

L 114-115 - Change "too large intakes" to "excessive intake".

L 119 - The beginning of this sentence is poorly-written. Change to "Since the three tested species are mainly frugivorous...".

L 128 - Add a comma after "test".

L 151 - Delete the "to".

L 159 - Change "that were not any more available" to ", which were no long available".

L 163 - Change "to" to "in".

L 173 - Change "on" to "as to".

L 179 - Delete "on".

L 192 - Change "because of" to "due to".

L 193 - Change "have been" to "were".

L 205 - Change "the each other position" to "one another".

L 256 - Change "fist" to "first".

L 258 - Change "have" to "had".

L 269 - Change "Since we" to "but".

L 277 - Change "tend" to "tended".

L 278 - Change "forage on" to "were foraging in".

L 280 - Change "previous" to "previously".

L 284 - Change "fine" to "fine-scale".

L 286 - Delete "than".

L 300-301 - Change "despotic degree" to "degree of despotism" and "potentially ultimately affecting" to "which could potentially affect their".

L 325 - Add "that" after "hypothesized".

L 326 - Change "being" to "may be".

L 328 - Change "if the variable species were as" to "whether each species was equally".

L 329 - Change "Like so" to "Thus".

L 338 - Change "were the individuals" to "they were".

L 340 - Change "had not" to "did not have the".

L 342 - The "it" here, or what is being tested, needs to be repeated for clarity.

L 350 - The reference needs a closing parentheses.

L 351 - I think "necessaries" needs to be changed to "necessary" here.

L 353 - Change "variables" to "variable".

L 361-362 - Change the end of this sentence to "sequentially removing non-significant terms of higher complexity".

L 365 - Add "a" after "we".

L 371- Pluralize "effect".

- L 375 – Change again to “sequentially removing non-significant terms of higher complexity”.
- L 379 – Here there is a double-negative. Do you mean “0 for no competitor present and 1 if at least one competitor was present”?
- L 380 – Pluralize “integer”.
- L 384 – Deleted “deeply”.
- L 385 – Change the middle of this sentence to, “did not seem to suffer from the reduced sample”.
- L 388 – Delete “the needed”.
- L 396 and 397 – I’m not sure I understand what “handed on” means in this context.
- L 408 – Change “of” to “from”.
- L 455 – Change “to notice” to “noticing”.
- L 471 – Change “when having faced” to “after choosing”.
- L 491 – The way this is written with the colon and the “they” reads as if M. tonkeana visited non-baited boxes 5-times more, which can’t be right...?
- L 504 – This would read better if a comma was put after “study” and “the” was deleted.
- L 508-509 – Delete “significantly more” and put “significantly more often” after “site”.
- L 510 – Change “with” to “there was”.
- L 512 – Add “rather” after “clumped”.
- L 514 and 558-561 – These results bring to mind a new paper on foraging site selection in lemurs which also found that more frugivorous species were more goal-directed and more insectivorous species are more exploratory, which is in line with your findings (Teichroeb & Vining, 2019, Navigation strategies in three nocturnal lemur species: Diet predicts heuristic use and degree of exploratory behavior. *Animal Cognition*).
- L 516 – Change “when” to “after”.
- L 523 – Change “showed” to “shown” and “travelling cost” to “travel costs”.
- L 528 – Add “and” before “planning”.
- L 562 – Please define what you mean by “opportunistic behavior” here.
- L 566 – Change “have” to “had a”.
- L 567-568 – Change “may pinpoint” to “suggest”.
- L 599 – Changed “faced” to “chose” and “change more” to “changed the distribution they were foraging in more”.
- L 603 – Add “in” after “help”.
- L 608 – Add “at” after “fed”.
- L 616 – Change “pursuit the” to “pursue”.
- L 617 – Add “a” after “and”.

Author's Response to Decision Letter for (RSOS-181722.R1)

See Appendix C.

Decision letter (RSOS-181722.R2)

08-Apr-2019

Dear Dr Trapanese,

I am pleased to inform you that your manuscript entitled "Where and What? Frugivory is associated with more efficient foraging in three semi-free ranging primate species" is now accepted for publication in Royal Society Open Science.

on behalf of Dr Atsushi Iriki (Associate Editor) and Professor Kevin Padian (Subject Editor)
openscience@royalsociety.org

Appendix A

Review of "Where or what? Primates in "miniature nature"?: frugivory triggers spatial cognition to forage efficiently"

Using semi-free ranging primates of three species, this study examined choices of food patches which differed in terms of distribution (clumped vs. scattered), quality (highly preferred vs. less preferred), and error rate of visiting (non-baited boxes). The authors found 1) clumped distribution was preferred by all species, 2) food quality had an effect on the choice, and 3) *M. tonkeana* had the most accurate spatial memory to discriminate between baited and non-baited boxes. The results of 2) and 3) support the authors' hypothesis that frugivorous species rely on spatial memory more than less-frugivorous ones. This is a well-controlled study using semi-free ranging populations to investigate the several ecological and cognitive factors affecting on their choices of food items.

General comments

With small number of subjects, the most robust finding of the study is that monkeys chose clumped over scattered distribution regardless of the difference of the degree of frugivory. However, the authors emphasized the *M. tonkeana*'s performance showing choices affected by quality of food (what), arguing better or stronger tendency of relying on the spatial memory in the more frugivorous species. I am not sure whether this type of argument is valid enough to relate their findings to spatial cognitive ability of three species using data on limited number of subjects with limited control conditions. One can argue that frugivory species perform well in the task using fruits as rewards because they are more motivated to collect them by nature. As authors admitted in the discussion, other factors like territory scale, opportunistic foraging, and motivation, would have been affected on their choices differently to each species. If the authors want to evaluate the pure spatial (or working) memory abilities among these species, they should use the captive subjects under more controlled environments. In my view, the important finding obtained in the study, however, is the observed interaction among several factors like food distribution, strength of food preference, social ranking, and learning ability. Because of this reason, I feel a bit strange on a phrase of the title, "frugivory triggers spatial cognition to forage efficiently".

Specific comments

Line 34 (Abstract)

The last phrase "have implications for the conservation of endangered species" is not appropriate for the abstract, because the issue is mentioned only in the last one sentence

in the discussion.

Line 99-100, 300-302

The small numbers of subjects from each species participated in the study make it difficult to evaluate the effects of social ranking.

Line 164-165, 401-403

The food items employed in the experiment were, according to the authors, palatable and had fairly comparable nutritional values. The fact that almost all food items are fruits would have been advantageous for more frugivory species because they are more motivated to participate in the task. If it is the case, the obtained results do not support the idea that more frugivory primates rely on spatial cognitive ability. Additionally, when you compare the several primates, it is necessary to consider other variables than calories in choosing the food items, because they have different digestive tracts with different time course of the digestion, affecting their motivation to intake.

Line 216-217, 406-408

The authors did not differentiate the spatial cognitive abilities involved in the task. But the cognitive function for “goal directed movements towards the baited boxes rather than visiting non-baited ones” is so called “working memory” which is supported by the brain areas (e.g. prefrontal cortex) different from those for purely spatial memory (e.g. hippocampus). They need to clarify which of the neurocognitive functions is corresponding to the task features in the study, because the difference in the ability of each species would explain a part of the difference in their performance.

Line 365

Missing a period? (travel “.” Sapajus sp. ...)

Line 385-386

Do the authors want to suggest that negative or stressful emotions in subordinate species would be associated to decreased spatial cognition? Is it contradictory to the fact that “subordinates visited fewer empty boxes than dominants” (line 320)?

Figure 2 and 3

“NS”s above the graphs can be limited to the important differences. It is hard to see the difference in the size of the plots (corresponding to the number of actions).

Appendix B

Dear Editor-in-Chief and Referees,

Thank you for your kind letter of "Decision on Manuscript ID RSOS-181722" on the 23th January 2019. We are very grateful to the reviewers' comments for the thoughtful suggestions. Based on these comments, we made careful modifications to the original manuscript, particularly we added a) another statistical model to strengthen our previous findings to meet reviewers' concerns and b) more rigorous statistical control accounting for multiple testing. Our main results were confirmed with these additional approaches. c) We better explained the complex experimental protocol and the methods. We realized that since our article was transferred directly from Proceeding of the Royal Society B (PRSB) to your journal, to meet the PRSB's length requirements a few parts of the methods were placed in the supplementary materials to the detriment, though, of the comprehension of the complex experimental protocol. d) Finally, we also ran simulations to explain in better details the rationale behind our statistical models, specifically to address the concerns of the reviewer 2. We believe that the manuscript has been greatly improved and we hope it has reached the standards required by your journal. We uploaded both a copy with track-changes and a clean one. As suggested by both reviewers, we modified the title of the manuscript as following: "Where and What? Frugivory is associated with more efficient foraging in three semi-free ranging primate species". Please find below our specific answers to the referees.

Yours sincerely,
Cinzia Trapanese

Referee 1

Review of "Where or what? Primates in "miniature nature"?: frugivory triggers spatial cognition to forage efficiently"

Using semi-free ranging primates of three species, this study examined choices of food patches which differed in terms of distribution (clumped vs. scattered), quality (highly preferred vs. less preferred), and error rate of visiting (non-baited boxes). The authors found 1) clumped distribution was preferred by all species, 2) food quality had an effect on the choice, and 3) *M. tonkeana* had the most accurate spatial memory to discriminate between baited and non-baited boxes. The results of 2) and 3) support the authors' hypothesis that frugivorous species rely on spatial memory more than less-frugivorous ones. This is a well-controlled study using semi-free ranging populations to investigate the several ecological and cognitive factors affecting on their choices of food items.

ANSWER: Thank you very much for this appreciation.

General comments

With small number of subjects, the most robust finding of the study is that monkeys chose clumped over scattered distribution regardless of the difference of the degree of frugivory. However, the authors emphasized the *M. tonkeana*'s performance showing

choices affected by quality of food (what), arguing better or stronger tendency of relying on the spatial memory in the more frugivorous species. I am not sure whether this type of argument is valid enough to relate their findings to spatial cognitive ability of three species using data on limited number of subjects with limited control conditions. One can argue that frugivory species perform well in the task using fruits as rewards because they are more motivated to collect them by nature. As authors admitted in the discussion, other factors like territory scale, opportunistic foraging, and motivation, would have been affected on their choices differently to each species. If the authors want to evaluate the pure spatial (or working) memory abilities among these species, they should use the captive subjects under more controlled environments. In my view, the important finding obtained in the study, however, is the observed interaction among several factors like food distribution, strength of food preference, social ranking, and learning ability. Because of this reason, I feel a bit strange on a phrase of the title, “frugivory triggers spatial cognition to forage efficiently”.

ANSWER: Thank you for this interesting comment. As mention in the article we are aware that other ecological or social factors can affect subjects' performances and we acknowledged for that in the manuscript in line 709 “Furthermore, other variables that we could not control for in our semi-free ranging protocol may have affected initial foraging choices of the individuals. First, first visited boxes may have been affected by the presence of competitors at the boxes of the non-chosen distribution. Overall, we did not find any effect of social competition on the foraging decisions even for the more despotic species (*M. fascicularis*; Table S1). However, in our experimental protocol we could test only for the influence of the number of individuals that were present around the chosen box since we could not collect information on the presence of (dominant) individuals at the non-chosen boxes, thus at the non-chosen fruit distribution. Therefore, assessing solely the first feeding choice in our study design where we could only partially test and control for social competition, it may not allow depicting true feeding decisions.”. We also acknowledge for the low sample size in terms of individuals tested per species. However, in the new version of the manuscript we further underlined these two points: please see for instance the line 123 “The relatively low sample size per species (N=14, Table S3) is because we tested only individuals i) with fully development of cognitive capacities, i.e. adults, ii) who voluntary took part at the Habituation and the Test Phase, and iii) who passed the criteria for the Habituation Phase (see paragraph “Habituation to the experimental protocol” in ESM)”, and the line 754 “However, given the small sample size in number of individuals tested per species, any generalization of the results at the species level are tentative, and further studies are needed to confirm the observed trend.”

Regarding the comment that the more frugivorous species had higher motivation thus performed better we do not totally agree because: a) the food preference test that we carried aimed at increase inter-species comparability of the behaviour allowing to choose the most preferred fruit types for each species. This is also underlined by the similar level of participation of tested individuals from the three tested species to our experiments. In addition, b) we did not observe differences in the species attraction towards each of their favorite fresh fruit during both the food preference test and the Test Phase. Last but not least, c) all tested species had a fairly high degree of frugivory (even the most omnivorous include a consistent percentage of fruit in their diet: *M. tonkeana* 71%, *M. fascicularis* 67%, *Sapajus* sp. 54%). Thus, we believe that the selected preferred fruit types were appetent food for all three tested species. In the revised manuscript, we better underlined these concepts for example in

line 136 “Being frugivory the main diet tendency of the three tested species (range of the percentage of fruit in their diet, between 54-71%, see Table S1 in ESM), fruit was an appetent food for all of them and no differences in the motivation to participate to the experiments were observed among the species.”. However, we acknowledged that the results of the Binomial Test in which the Tonkean macaques are the only individuals who after feeding from the clumped fruit, fed above chance level also from the scattered fruit, could be likely also explained by higher motivation to pursuit the foraging on fruit.

Finally, this project is part of a larger project that includes a study where the same individuals will be also tested on similar tasks in more controlled environments (including also digital computerized tasks) to better assess their spatial memory abilities. Regarding the title we agree to modify it as following: “Where and What? Frugivory is associated with more efficient foraging in three semi-free ranging primate species”.

Specific comments

Line 34 (Abstract)

The last phrase “have implications for the conservation of endangered species” is not appropriate for the abstract, because the issue is mentioned only in the last one sentence in the discussion.

ANSWER: We agree with this and we deleted it from the abstract.

Line 99-100, 300-302

The small numbers of subjects from each species participated in the study make it difficult to evaluate the effects of social ranking.

ANSWER: We agree with it and in addition, when running again the models while controlling for multiple testing, we did not find a significant effect of “rank” any more in any model: see line 613 “Rank did not affect the overall visiting “error rate” of baited boxes ($\chi^2=2.183$, $df=1$, $p=0.139$)”, and line 571, Although species had no effect in the presented model ($\chi^2=3.200$, $df=2$, $p=0.202$), the analysis revealed potentially significant effect of species in paired interaction with task/season ($\chi^2=7.155$, $df=2$, $p=0.028$), the number of competitors at the box ($\chi^2=9.096$, $df=2$, $p=0.011$) and the distribution of the box ($\chi^2=8.429$, $df=2$, $p=0.015$), respectively. However, the control procedure for maintaining the false discovery rate at 5% (see paragraph “Models’ description”) revealed that those aforementioned paired interactions are likely false positive, therefore they were discarded from the model and we considered independent variables for interpretability and limitation of overparameterisation [52]”.

Line 164-165, 401-403

The food items employed in the experiment were, according to the authors, palatable and had fairly comparable nutritional values. The fact that almost all food items are fruits would have been advantageous for more frugivory species because they are more motivated to participate in the task. If it is the case, the obtained results do not support the idea that more frugivory primates rely on spatial cognitive ability. Additionally, when you compare the several primates, it is necessary to consider other variables than calories in choosing the food items, because they have different digestive tracts with

different time course of the digestion, affecting their motivation to intake.

ANSWER: We see your point and we totally agree that different species have different physiological variables in terms of gut passage rate, digestion ability, metabolism rate etc.. For instance, given the smaller body size, capuchin metabolism is likely faster than the one of macaques: this may translate in a higher attraction for different food types. In the food preference test we used the same food types for the three species since a) all tested species are mainly frugivorous (between 54-71% of fruit in the diet), and 2) to use as much as possible the same protocol to allow inter-species comparison. From our observations both during the food preference test and the experiments, variation in motivation was higher intra-species than inter-species. To account for this, we treated the variable “subjects” as random factor in our GLMMs. Moreover, as mentioned in the manuscript, all tested species are usually provisioned with fruit once per week, and they are all very attracted by fruit. During the whole study period we suspended the fruit provisioning for the three species. In the revised manuscript we moved this information from the supplementary materials to the main text to make them clearer: please see lines 131-133 “However, the supply of fruit (once per week) was suspended during the study period to increase primate motivation and to avoid too large intakes because of the experiments (see EMS “Food preference test”).”. Please see also lines 138-141 “Being frugivory the main diet tendency of all tested species (even the most omnivorous include a consistent percentage of fruit in their diet, between 54-71%, see Table S1 in ESM), fruit was an appetent food for all of them and no differences in the motivation to participate to the experiments were observed among the species”.

We believe that our experimental protocol (particularly the specie-specific food preference test together with a fairly high frugivory level of each species) helped to minimize eventual differences in motivational level among tested species. Therefore, for the aforementioned reasons, we do not think that our results of interspecies comparison were affected by different motivational level among the tested species.

Line 216-217, 406-408

The authors did not differentiate the spatial cognitive abilities involved in the task. But the cognitive function for “goal directed movements towards the baited boxes rather than visiting non-baited ones” is so called “working memory” which is supported by the brain areas (e.g. prefrontal cortex) different from those for purely spatial memory (e.g. hippocampus). They need to clarify which of the neurocognitive functions is corresponding to the task features in the study, because the difference in the ability of each species would explain a part of the difference in their performance.

ANSWER: Thank you for bringing up this tricky point. The main cognitive ability investigated in our study is “spatial memory”, since each season lasted five days during which subjects were tested twice per day. Thus, once at first they remembered the positions of the baited boxes available for that task/season, then they needed to remember each day the positions of always the same baited boxes. Even if “working memory” is involved too (since at the first trial of each task/season (*when*), each subject had to remember the locations of the new fruit type available), we did not investigate this ability in this article. In the revised manuscript we made these points clearer adding the following sentence in line 625: “In this study we investigated the spatial memory ability, particularly related to spatial food distribution, since each task/season lasted five days during which subjects were tested twice per day”.

Line 365

Missing a period? (travel “.” Sapajus sp. ...)

ANSWER: It is true, thank you for noticing that.

Line 385-386

Do the authors want to suggest that negative or stressful emotions in subordinate species would be associated to decreased spatial cognition? Is it contradictory to the fact that “subordinates visited fewer empty boxes than dominants” (line 320)?

ANSWER: Thank you for pointing out that point. Given this confusion and the fact that after controlling for multiple testing the results did not revealed anymore a significant difference between subordinates and dominants’ performances, we removed this sentence in the revised manuscript.

Figure 2 and 3

“NS”s above the graphs can be limited to the important differences. It is hard to see the difference in the size of the plots (corresponding to the number of actions).

ANSWER: Since we added to the manuscript a new model (“Change of the distribution”) and the relative figure to address the concerns of the reviewer 2, we removed the abovementioned figure relative to model “Successive five boxes CvQ”.

Reviewer: 2

Comments to the Author(s)

In this paper, the authors compare the foraging decisions of three species of captive primates, two macaques and one capuchin species, on two food arrays, one clumped and one dispersed, baited with foods of different preference. They use this set up to mimic foraging in a seasonal environment and they also looked at the degree that travel was goal-directed within the experiment. They interpret their results as showing that all three species preferred the clumped array. However, this preference was not as strong for the most frugivorous species when preferred fruit was scattered. This species also used more goal-directed travel. The authors interpret this to show that the most frugivorous species was the most efficient foraging taking into consideration what food was where when making decisions.

There seems to be valuable data here but the description of the experiments and the presentation of the results are often confusing and it is difficult to understand exactly what has been done.

ANSWER: Thank you for this comment. We agree that the protocol is complicated, and it was confusing because a large part of information was placed in the supplementary materials. In the revised manuscript we thus moved details of the protocol and the methods in the main text. We hope that now everything is much clearer. Here some examples of added lines into the main text: line 152 “Each box was consistently filled with the same fruit provided in pieces in the same quantity (two pieces) and size (see ESM for details). Therefore, the location of each box was

associated solely to a given fruit type throughout the experiment.” Lines 198-212 “To avoid giving them cues on which boxes were available (baited) in a given trial, we pretended to fill also boxes of other the tasks/seasons (randomised at each trial) while filling the boxes of the current task/season (six more boxes were filled for *M. tonkeana* and 12 for *M. fascicularis* and *Sapajus* sp.; the difference is due to a protocol improvement after testing the *M. tonkeana*; see ESM “Test Phase” for details). At each trial of a task/season the experimenter waited at the starting point for the passage of a possible focal individual. When a focal subject crossed on the starting point, the experimenter showed to him/her his/her personal starting cue while keeping a neutral face expression and body posture (i.e. always orientated towards the centre of the outdoor area). This allowed us to avoid delivering unintentional cues to the focal individual about the position of the nearest baited box. When the focal individual moved, the experimenter followed him/her within 2 meters (focal animal sampling: [44]). Each time the individual stopped, the experimenter remained still, oriented towards the subject direction and waiting for the subject to move again”.

I also have some serious issues with the interpretation of the Results that I will detail below. I am not sure that the authors actually found what they say they found.

As a first point, I really think that the title needs revision. Most people will not understand right away what is meant by “miniature nature”. Are these animals semi-free ranging? Or should they be considered captive? I would leave this out of the title. The word “triggers” is also contentious in the title. It implies causation but this is not shown by the data. A better title would be something like, “Where and What? Greater frugivory is associated with more efficient foraging in three semi-free ranging monkey species”.

ANSWER: We agree and we changed the title as suggested: *Where and What? Frugivory is associated with more efficient foraging in three semi-free ranging primate species*

Early in the Introduction of the species tested, some information should be included on the social organization and dominance regimes of these species. Later in the discussion some conclusions are presented based on which species is more despotic and which is more tolerant but these features are not introduced.

ANSWER: Thanks for this suggestion, we agree and thus we added this information in the introduction (lines 89-92): “while the main dietary tendency of the three species is frugivory, *M. tonkeana* are more frugivorous and tolerant [35;36] in comparison to *M. fascicularis* that are frugivorous/omnivorous and despotic [37;36] and *Sapajus* sp. [38;39] that are even more omnivorous but socially quite tolerant”.

At the end of the Introduction, it would be helpful if what was done was clarified more. For instance, on line 81, a “miniature nature protocol” should be defined. Later, on line 84 and below, it needs to be stated whether animals were tested alone or in groups. If in groups, how was the competition with other group members accounted for? In addition, the reasoning behind the prediction made on lines 86-89 should be presented. Why was the more omnivorous species not expected to feed on the scattered, preferred food first?

ANSWER: We acknowledge that those concepts were not clear since they were placed in the supplementary materials. We thus moved several sentences into the main text:

- We deleted the definition “nature miniature protocol” to avoid confusions (line 96): “We investigated how primate foraging decisions were affected by clumped and scattered food distributions and food preference (highly vs. less preferred, a proxy for food quality) by controlling and manipulating the variables *where* (food distribution) and *what* (fruit availability) in this semi-natural context”.
- Animals were tested individually but in their group, as stated in the methods in line 145: “We used a remote control system to unlock visited boxes from a distance, rendering the fruit available only when a focal subject [44] touched a box. This allowed us to test subjects individually within their social group.”. We believe that it more appropriate to leave this sentence in the Methods part rather than in the Introduction.
- We made our prediction less assertive and clearer. We predict that more generalist primates, the more omnivorous species, may not react similarly to the more frugivorous species to the change in fruit quality and distribution. We may expect that the more frugivorous primates may be more finely affected in their foraging decisions by the changes in fruit quality in the scattered and clumped distribution. However, to make this concept clear (but keeping it short) we modified the prediction as following (line 102): “Since fruit is an ephemeral and scattered resource, if fruit quality (in terms of preference) is was also a major driving factor affecting their spatial foraging decisions, we predicted that when the most preferred fruit (what) was scattered, the more frugivorous primates, the *Macaca* spp. would be more affected in their foraging decisions in comparison to the more generalist (omnivorous) species *Sapajus* sp. (Table S1), than when fruit quality do not vary between the two distributions. When the most preferred fruit (what) is scattered, we predict that more frugivorous primates, the *Macaca* spp. (Table S1), feed first on this before feeding on less preferred fruit in a clumped distribution, in comparison to the more omnivorous species *Sapajus* sp. Since fruit is an ephemeral and scattered resource, Since fruit is an ephemeral and scattered resource, the more frugivorous species were expected to rely more on the spatial memory of the food distribution and to have more goal-directed travel towards the each feeding sites in comparison to the more omnivorous species [41, 43]. We hope that now it is clearer.

In the Methods, the description of the models that were run is very confusing. Especially where interaction terms and comparisons of models up. I think this could be rectified by amalgamating all the information provided in the Description of Statistical Models section (L 230-253) into the introduction of each model earlier. If each model were discussed fully, in turn, it may help the reader significantly.

ANSWER: We apologize once again for the confusing description of the models. We made the suggested changes to increase the understanding of the models by unifying all model information in one paragraph. In addition, to make the statistical analysis clearer we also well separated each part in different paragraphs: “Assessing the preference for the clumped or scattered distribution: Index of Lateralisation”, “Assessing reasons for a preference for a fruit distribution and the strategies used to forage: Generalised Linear Mixed Models”, “Models’ description”, “Models implementation”. We hope that now this part is clearer and easier to understand.

Another point that would improve the readers understanding of what was done would

be to provide Figures S6a and b and S7 within the main manuscript rather than in the ESM. These are vital to understanding the set up and the reader should not have to search for them.

ANSWER: Thanks a lot for suggesting that. We completely agree and included now these figures in the main manuscript (Figure 1a, 1b and 1c). We added the scale for each figure and we precised the distances among the boxes in the main text (line 232): “Each distribution was composed of six boxes, one with a clumped configuration (range distance between boxes 2-3m) and the other with a more scattered arrangement (range distance between boxes 15-18m). Because of the dense vegetation, the boxes of the same task/season were not visible from the each other position.”

My main issue with the interpretation of the Results is that I am not convinced that the monkeys actually preferred the clumped distribution, which is a key finding from the study. The first visited box was not in the clumped distribution significantly more often. Even though the five successively visited boxes were more likely to be from the clumped distribution if the first choice was from that distribution, this does not tell us anything. Once the animals had made their initial decision, it is not surprising that they then chose food sites that were nearby. There is a large set of literature showing distance minimizing in primates and other animals while foraging. In Fig. S6a and b and S7, it seems that sites within each distribution are closer to one another than the distance of the distributions from one another (exact distances should be provided). A model could be run to show whether, when choosing the scattered distribution first, the monkeys then stayed within that distribution for the next successive five choices. That could perhaps tell you if they foraged to decrease the distance traveled between sites (which is still somewhat interesting), even if they did not make the best choice initially. The authors may have to change the focus of the study slightly or find another way to convince the reader that the clumped distribution was actually chosen more often.

ANSWER: Thank you for bringing up this point and for suggesting a clear model. Indeed, to our understanding the model you suggested to run corresponds to what we did in the models “Successive five visited boxes CvS” and “Successive five visited boxes CvQ”. Probably this misunderstanding is due to our lack of clarity in methods’ section. In fact in the models “Successive five visited boxes CvS” and “Successive five visited boxes CvQ”, we aimed to investigate the foraging strategy used by primates after the first chosen box: we assessed the distribution of the second to the sixth box chosen (i.e. from the clumped or the scattered distribution). Thus, we tested whether individuals targeted the closest neighbouring box of the same distribution or if they moved to the opposite distribution after choosing a box of the clumped or to the scattered distribution. We found that the probability of selecting a box from the clumped distribution was higher if the previously visited box was from the clumped distribution. However, in the revised manuscript, to increase strength and clarity of our results we ran an additional model (“Change of the distribution”) aiming at assessing the “reason of the preference” for one or the other distribution. Specifically, we tested if the propensity to stay in a given distribution or to leave it depended on 1) whether the previous visited box was baited or not, 2) its location, whether this box belonged to the clumped or the scattered distribution, 3) the tested species, 4) the task/season and 5) the rank of the focal subject (see lines 315-326). This additional model showed that the probability to change distribution was higher if primates were the chosen previous box was a non-baited one (compared to baited ones, see line 580). These results aid to explain the non-significant results for the clumped distribution in the models “First visited box”, and to make a clearer link between the results of the “First box visited”

and the “Successive five visited boxes”. Likely, when individuals chose at first a box from the more scattered distribution, they then corrected themselves at the successive choice to feed on the more profitable clumped distribution (to maximize energy spent/meters covered). It may be possible that primates possess a finer scale (local) spatial memory of food distribution rather a more global spatial memory of the area. We made all those points clearer in the discussion (lines 692-742): “In general, the results of the model “Change of the distribution” show however that all three species have higher probability to change distribution, thus re-correct themselves, after choosing a non-baited box. This result may pinpoint that primates readjust their strategy based on trial-error processes. This behaviour may help to explain why we did not find significant results from the model “First box visited” while we found a general preference for the clumped distribution with the Index of Lateralisation. Indeed, it could be that primates still memorised the working scenarios with the underlying idea of two existing fruit patches in which foraging costs differ: one with potentially more distant boxes, and therefore error-prone, or another with closer boxes. If so, making a mistake could be an indication of being in the more error-prone patch and that signal would be processed and analysed, resulting in a change of foraging distribution. However, this is logical only if primates could at least empirically determine differences among patches in the local probability of finding a non-baited box. Here, we observed that monkeys did not make errors consistently across distributions and context: in CvS1 or CvQ they failed more in finding baited-box in the scattered distribution, but this was the opposite in CvS2 (see model “Goal-directed strategy”). Therefore, it leaves open the questions of the cognitive mechanisms that are at play in primates foraging decision-making, especially with regards to orientation/re-orientation events. Furthermore, other variables that we could not control for in our semi-free ranging protocol may have affected initial foraging choices of the individuals. First, first visited boxes may have been affected by the presence of competitors at the boxes of the non-chosen distribution. Overall, we did not find any effect of social competition on the foraging decisions even for the more despotic species (*M. fascicularis*; Table S1). However, in our experimental protocol we could test only for the influence of the number of individuals that were present around the chosen box since we could not collect information on the presence of (dominant) individuals at the non-chosen boxes, thus at the non-chosen fruit distribution. Therefore, assessing solely the first feeding choice in our study design where we could only partially test and control for social competition, it may not allow depicting true feeding decisions.

Overall, the effect of social competition on the foraging decisions was more evident in the more despotic species (*M. fascicularis*) than in the other two species with more tolerant societies (Table S1). The preference for a specific side of the outdoor area found in two subordinate individuals can also be linked to social competition: subordinates avoided taking risks in the most frequented part of the area. Moreover, negative or stressful emotions are associated to decreased spatial cognition in humans [66]. While considering these issues, our results from the Index of Lateralisation and the GLMMs suggest that primates remembered the position of the baited boxes and of the most profitable distribution. It can be argued that this general preference towards the clumped distribution could have been due to our experimental settings that provided a local difference in baited box density once the subject chose to forage on a given distribution: i.e. higher density of baited boxes within the clumped distribution in comparison to the scattered one (see map of the outdoor experiments, Figure 1a,b,c). If such, we would have expected strong differences in the tendency to change distribution, with this probability being much higher for changes from the scattered to

the clumped one than the reverse. Our results show that, if individuals faced a non-baited box, they then change more distribution than when opening a baited box (model “Change of the distribution”). It is important to note that the distribution of the previous box had no clearly significant effect in interaction with the nature of the box (i.e. baited or not), nor independently (both were identified as highly likely false positive). These results therefore help interpreting the overall preference for the clumped distribution assessed with the Index of Lateralisation, since it helps excluding the hypothesis of a local density bias of boxes as the distribution has weak to no effect. However, it leaves still open the questions of the cognitive mechanisms that are involved in primate foraging decision-making, especially with regards to orientation/re-orientation events.”.

Finally, as it seems that we missed the shot in explaining how the model “Successive five boxes” could be used to analyse the strategy adopted by individuals, we would like to provide here below also theoretical expectations framed via simulation within the paradoxical situation of no first box preference, but overall preference. We hope that it will help increasing clarity on the rationale behind our statistical models and to show how these can lead us to the made conclusions:

In the following part, S_t represents the choice made at step t , namely 0 or 1, whether the animal chose to forage on one distribution or the other one. Here, for having the same referential, 1 represents situations in which the animal foraged on the clumped distribution, and 0 on the scattered one.

In the frame of our experimental setting, we hypothesized that monkeys could follow any of the following strategy:

- Foraging exclusively, or mostly, on one of the distributions (i.e. permanent bias, Scenario 1)
- Foraging on both distributions, with S_{t+1} being conditional to S_t independently of the value of t (i.e. of the location of the step within the step series)
- Foraging on both distributions, with S_{t+1} being conditional to S_t and depending on the value of t (i.e. of the location of the step within the step series)

To analyze whether our approach could capture those different behaviours, we simulated data for six individuals that proceeded to 250 trials as described in the article, and who followed consistently across all trials the aforementioned behaviours. To do so, we modelled each decision step, in which the monkey either decides to continue foraging on the distribution it foraged before or moves to the opposite distribution. That consisted in a Bernoulli experiment, with p , classically representing the probability of a success, here corresponding to foraging on the clumped distribution. A permanent bias towards a distribution then means p being invariant across time, and strictly superior to 0.5 if the bias is towards the clumped distribution, else strictly inferior if the bias is towards the scattered distribution. Here, we used $p=1$, $p=0.7$ and $p=0.3$ (Scenario 1a, 1b, and 1c). A conditional behaviour, but independent of the time, was modelled with p taking different values based on the previous step. As such, we hypothesized that the individual could either consistently, or mostly, choose to continue foraging at S_{t+1} on the same distribution it foraged at S_t (Scenario 2a with $p=1$ if $S_{t+1}=1$, else 0, and Scenario 2b with $p=0.7$ if $S_{t+1}=1$, else $p=0.3$, therefore corresponding to a variation in intensity of this strategy) or the opposite (Scenario 2c, with $p=0$ if $S_{t+1}=1$, Scenario 2d, with $p=0.3$ if $S_{t+1}=1$, else $p=0.7$). Note that, in our case, the situation is symmetrical between the two distributions, meaning that no matter where the monkey is, the probability to switch to

the opposite distribution remains the same. The last behaviour consisted in foraging on the same patch for the first three steps, and then in foraging on the opposite distribution for the last three steps. For each scenario, we randomly attributed a rank to the individuals and a number of competitors. In addition, we maintained the premise that individuals chose the first box at random, as shown with our analysis, in order to demonstrate that despite this, differences can be made between strategies and are not uninformative as suggested. We analysed the output of the simulations with general linearized model (as described in our methods). However, here, we used simpler models not considering individual random effects, and limited the model to three variables: the previous choice, the number of competitors at the box and the rank of the individual, the latter two being randomly modelled to have on average a null effect here, and only serves as control when testing for a full vs. null difference. We repeated this complete procedure 1000 times to look for broad pattern emerging from these analyses.

Results:

This shows that the output pattern clearly differentiates between each strategy. Indeed, in the absence of an initial differential choice for the first step (i.e. random choice only at this step), the overall lateralisation index indeed displays if there is an overall bias (i.e. distribution preference), as only in all sub-cases of Scenario 1 individuals were largely lateralised, while the range of lateralisation of the individuals clearly represent the overall populational bias it exists (Figure 1).

Figure 1: Lateralisation rate in function of the modelled scenario (general preference, conditional choice with time independency, conditional choice with time dependency).

On the opposite, the linear model lets us characterize the conditionality of the choice: only in Scenario 2 and 3 do evidence null vs. full differences due to the significant impact of the previous choice (Figure 2).

Figure 2: Rate of significance observation of the full vs. null model, of the intercept, and of the variable previous choice, in function of the modelled scenario (general preference, conditional choice with time independency, conditional choice with time dependency).

It even allows us to further refine the comprehension of the decisional process that is at play. As depicted by Figure 3, the direction of the estimates of the variable “Previous choice” are true indicators of whether there is positive feedback loop (i.e. the individual tends to remain where it was the step before), or negative feedback loop (i.e. the individual tends to avoid remaining where it was the step before), with the absolute value being directly related to the robustness of this choice (i.e. here, the value of p). That last point is of true importance as it allows us quantifying the value of p, respectively when the previous box chosen one from one given distribution, as indeed, although we considered symmetrical behaviour (i.e. same conditional probability),

those might differentiate according to the distribution individuals foraged before (see our results in the article). Finally, by combining the lateralisation index and the linear model, we can differentiate Scenario 2a or b, with Scenario 3: both are identified with a significant positive effect of the previous choice, but only in Scenario 2 do we observe lateralisation.

Figure 3: Estimate value of the intercept or of the variable previous choice (ref is foraging on the scattered distribution, i.e. “0” in our Bernoulli experiment) in function of the modelled scenario (general preference, conditional choice with time independency, conditional choice with time dependency).

Comparison to our results:

In the results of our experimental study, we do see overall lateralisation, which clearly shows that there is a populational bias towards the clumped distribution. As this distribution varied in its location overtime, we therefore take it as a potential evidence that individual favoured voluntarily the clumped distribution to forage the first six boxes, despite the absence for a preference at the first step. However, we also observed a significant effect of the previous choice: more particularly, we do see that subjects tend to favour remaining in the distribution they are in (which, as you said, is unsurprising although we might argue that it is possible to emphasize that animals might adopt a back-and-forth strategy at first to clearly identify their environment, only taking the

decision to stay at a given patch afterwards), but, and we believe this is more meaningful, the situation is not symmetrical: individuals tended to leave a bit more frequently the scattered distribution to forage on the clumped one than the reverse, and this regardless the species. This observation is depicted in the Figure 3 of the article, which underlines that the probability of selecting a box from the clumped distribution when being in the scattered distribution is not one minus the probability of selecting a box from the clumped distribution when being in the clumped distribution. This is clearly consistent with the result of a general bias (see explanation above), but therefore provides the additional information that subjects might correct their foraging strategy when realising that they do not face the optimal situation between those that they know. In other words, it might indicate individuals consciously know if a given situation is of advantage as compared to another (foraging in a clump distribution rather than in scattered one), and therefore, could eventually let us think that foraging strategy are cognitively driven, although we sincerely acknowledge that further investigations are needed in line 1086 “However, given the small sample size in number of individuals tested per species, any generalization of the results at the species level are tentative, and further studies are needed to confirm the observed trend.”

Minor comments:

L 20-21 – It would sound better if the first two sentences were linked by adding an “and” after “challenging”. Also, add “have” after “studies” and change “to” to “with”.

ANSWER: Thank you for this suggestion, modifications have been made. The sentence is now reading as following (line 21): “Foraging in seasonal environments can be cognitively challenging and comparative studies have associated brain size with a frugivorous diet.”

L 24 – Was the *Sapajus* sp. not known or what this a mix of animals from different species? It would good if this could be clarified later in the Methods.

ANSWER: It is a hybrid species indeed. This is now clarified in the Methods, line 120: “*Sapajus* sp. was like a hybrid of the species *Sapajus apella* and *Sapajus nigritus*.”.

L 28 – Change “Primates” to “All species”, otherwise it is not clear whether species or individuals are being discussed.

ANSWER: Thank you for this suggestion, this has been changed.

L 28 – Change “with” to “but”.

L 29 – Add “did so” before “at”.

ANSWER: Thanks a lot, we made both these changes.

L 30 – Delete “using spatial memory” here since this is a supposition.

L 33 – Change “the environment” to be more specific. Maybe to something like “food type and distribution”.

ANSWER: Both those points were fixed as suggested.

L 34 – It is unclear how these results impact conservation. Please be more explicit or delete this.

ANSWER: We deleted this sentence from the abstract.

L 38 – Change “arduousness” to “difficulties involved”.

L 39 – Change “might be due to the” to “vary depending on qualities like a”.

L 40 – Change “the” to “a”.

The three suggested changes were made.

L 41-42 – These paragraphs should be joined.

ANSWER: Thank you for this suggestion, we fix it pooling the paragraphs as following (line 47): “To maximise foraging efficiency, animals should attempt to increase energetic intake and minimise costs”.

L 42-43 – This sentence is confusing. Please rewrite as, “To maximize foraging efficiency, animals should attempt to increase energetic intake and minimize costs”.

ANSWER: Thanks for your suggestion. The sentence was replaced with the sentence you suggested.

L 44 – Change “tropical forests” to “seasonal environments”. After all, truly tropical forests are notable for their lack of seasonality.

L 44 – “this dense environment” to “forests”.

L 45 – Delete “seasonal and” since it is redundant.

ANSWER: Thank you, we took into account those three suggestions and changed the text accordingly.

L 48 – Change “characterized by a” to “mean there is”.

L 49 – Change the first “and” to “with”.

L 52 – Start this sentence with “Evolution of the ability”.

L 57 – Add “have” after “mammals”.

ANSWER: We changed the text as suggested in those four comments.

L 58-59 – Move the second “fruit” to after “patchy” and change “phenology” to “availability”.

L 60 – Change “constantly” to “consistently” and add “and” before “hence”.

L 61 – Delete “noticeable”.

We changed the text as suggested in those three comments

L 65 – For clarity, consider changing the middle of this sentence to, “in shaping the evolution of cognitive complexity has gained precedence over the view...”.

ANSWER: This was changed as suggested.

L 73 – Add “to” after “planning”.

L 74 – Change “revisit” to “revisiting”.

L 77 – Change the middle of the sentence to, “allow the manipulation of variables...”.

L 78 – Add “the” after “compared” and “strategies” after “foraging”.

L 79 – Change “different” to “differing”.

ANSWER: Thank you, all these five suggestions have been considered.

L 81 – Please define what a “miniature nature” protocol is.

ANSWER: Thank you for pointing out that. Since it was confusing, we removed it from the whole text.

L 83 – Change “this” to “the”.

L 84 – Change “is” to “was”, change “and” to “versus”, and change “predict” to “predicted”.

L 85 – Add “would” after “primates”.

L 86 – Change “is” to “was”. These types of predictions are usually best in past tense throughout, since the work is already done.

ANSWER: Definitely in agreement, thank you. We changed all those four points in the manuscript.

L 87 - Change “is” to “was” and change “predict” to “predicted”.

L 88 – Add “would” before “feed” and “resource” after “this”.

L 90 – Change “are” to “were”.

ANSWER: Thank you, we changed these three points as suggested.

L 91 – What does “feeding sites” here refer to? Each food box? Or the overall array?

ANSWER: It refers to each single box. Thank you for this comment, this is now better specified (lines 113-114: “to have more goal-directed travel towards each feeding site in comparison to the more omnivorous species”).

L 102 – Change “spontaneous” to “natural”.

L 103 – Move “daily” to after “fed”, pluralized “pellet”, and change “provision” to “provisioning”.

ANSWER: Thank you, we changed both points.

L 104 – The part here “other fruit types have been used in experiments” is not clear. Do you mean that you did not want to over-feed them or was this to encourage participation in the experiments? Or both? Please clarify.

ANSWER: Thanks for pointing out this. As mentioned before, we did it to increase their motivation in taking part into the experiments. We now better specify this in the text (line 131 “However, the supply of fruit (once per week) was suspended during the study period to increase primate motivation and to avoid too large intakes because of the experiments (see EMS “Food preference test”).”).

L 106 – Delete “to be” and change the second “the” to “these”.

L 107 – Delete “to” and add “with” after “individuals”.

L 112 – Add “a” after “from”.

ANSWER: Thank you for these suggestions, we made all three changes.

L 114 – Since all boxes remained in the outdoor areas even when they were not baited, please clarify here how the monkeys were to know what boxes were baited. Was this based on memory of a previous trial? This needs to be explicit.

ANSWER: Thank you for pointing out this. Yes, they should have known based on memory of previous trials. The subjects learnt what boxes were baited in each task/season during the habituation phase; to clarify it, we added this sentence (lines 178-186): “The subjects were introduced to the spatio-temporal availability of different fruits by adding the boxes of the specific fruit to the outdoor area week after week, following the task/season order in the Test Phase (see ESM for more details). Thus, during the first week only the boxes of the first task/season were placed in the outdoor area. At the first day of the second week, the boxes of the second task/season were added to those of the first task/season that were not any more available (neither baited nor openable). Progressively, also the boxes of the third and then of the fourth task/season were added in the same way. After the first set of four weeks (i.e. The habituation in the outdoor area) all boxes were always present in the outdoor area.”.

L 119 – What does “(two)” refer to? To fruits or two sites?

ANSWER: It means two pieces of fruit. Thank you for pointing out this confusion, the sentence has been clarified (lines 152-154): “Each box was consistently filled with the same fruit provided in pieces in the same quantity (two pieces) and size (see ESM for details). Therefore, the location of each box was associated solely to a given fruit type throughout the experiment.”.

L 119 – Also please be explicit here as to why you pretended to fill 12 boxes. What were the monkeys supposed to get from this?

ANSWER: Thank you for highlighting this point. This is a crucial point: if monkeys would have seen us filling up only the boxes of that task/season they would have not needed to rely to their spatial memory to retrieve the baited boxes. This reason was specified just a bit below this sentence (lines 198-212): “To avoid giving them cues on which boxes were available (baited) in a given trial, we pretended to fill also boxes of other the tasks/seasons (randomised at each trial) while filling the boxes of the current task/season (six more boxes were filled for *M. tonkeana* and 12 for *M. fascicularis* and *Sapajus* sp.; the difference is due to a protocol improvement after testing the *M. tonkeana*; see ESM “Test Phase” for details)”. Given that this was unclear we specified it moving this sentence from ESM to the main text.

L 143 – So the experiment took 16-weeks total. This information should be included here.

ANSWER: Thank you for this suggestion, the information has been added (line 192).

L 145 – Delete “the primate”.

ANSWER: Thank you, this has been done.

L 149 – It says here the monkeys were “closely followed”. Please state how close and how you made sure that this did not influence the decisions and behavior of the animals.

ANSWER: Detailed information that answered to this comment were present in the supplementary materials. We moved them from there to the main text in order to increase clarity about our protocol, lines 204-212 ” At each trial of a task/season the experimenter waited at the starting point for the passage of a possible focal individual. When a focal subject crossed on the starting point, the experimenter showed to him/her his/her personal starting cue while keeping a neutral face expression and body posture (i.e. always orientated towards the centre of the outdoor area). This allowed us to avoid

delivering unintentional cues to the focal individual about the position of the nearest baited box. When the focal individual moved, the experimenter followed him/her within 2 meters (focal animal sampling: [44]). Each time the individual stopped, the experimenter remained still, oriented towards the subject direction and waiting for the subject to move again". In addition, together with the manuscript we provided a video of the focal following to increase understanding of our protocols and any reader concerns about the eventual impact of the observer.

L 154 – What does the “red line” refer to? Since you haven’t defined it yet, please put “(see below)” or something equivalent here.

ANSWER: Thank you for this comment. The sentence has been moved before in order to explain what the starting point was (lines 233-236): “The outdoor area was virtually split into two parts by a line whose points were equidistant from the closest box of each circular distribution (between 15-20m) and served as starting point for the trials (see the red line in Figure 1a, b, c)”.

L 156 – Add “to” after “way”.

L 158 – Change “this” to “competition by”.

ANSWER: Thank you, these two changes were made.

L 172-173 – This sentence should be broken up and rearranged to be clearer. “In CvS2, the relative positions of the clumped and scattered distributions was inverted.” Then describe the fruit that was available where. This part is very confusing. Does “similarly preferred” mean the same? And what do you mean by “only available in other boxes”?

ANSWER: Thank you for this comment, we apologize since the sentence was not clear. We modified it as following (line 240): “In CvS2, the relative positions of the clumped and scattered distributions were inverted: if the clumped distribution was on the East side in CvS1, it was on the West side in CvS2 (Figure 1a, b). The boxes were filled with a different fruit that was similarly preferred to the one used in CvS1 (Table S6)”.

L 184 – Please provide a formula for the Index of lateralization.

ANSWER: We now provided the formula at line 259: “We did so adapting the Handedness Index $HI=(R-L)/(R+L)$ (e.g. [45]), with R indicating the number of box choices in the right side of the outdoor area (i.e. the clumped distribution in CvS1 and to the scattered in CvS2) and L indicating the number of choices in the left side of the outdoor area (i.e. the scattered distribution in CvS1 and the clumped in CvS2; see Figure 1a, b, c)”.

L 186 – Change “area” to “enclosure”.

L 194 – Change “consisted in” to “was”.

L 206 – Start a new paragraph here.

ANSWER: Thank you for these three suggestions, we changed the text accordingly.

L 208 – Does similar spatial configuration refer to the same one? Not clear.

ANSWER: Thank you for this comment. The sentence “we considered the two tasks/seasons with a similar spatial configuration” means that we pooled together the data of the two tasks having the same configuration of the position of the clumped and scattered distribution. However we tried to clarify it (line 302) “To disentangle the

effect of the distribution (where) and the quality of fruit (what) on such a choice, we conducted the analysis separately: first on CvS1 and CvS2 tasks/seasons (Nvisited boxes=508), and then on CvS1 and CvQ tasks/seasons (Nvisited boxes=478) in which the distributions were identically distributed in the outdoor area, but fruit quality differed (Figure 1a, b, c; Table S6).”

L 212-215 – The description of the models is so confusing.

ANSWER: We acknowledge for this and as mentioned before in answer to other comments we added much more information on the models in different paragraphs (from line 254 to 486). We hope it is now easier to understand.

L 217 – I still don't understand how the monkeys were supposed to know which boxes were unbaited. I see that this is stated in the ESM but a brief statement at the end of the Introduction in the paper is also needed.

ANSWER: Thank you for this comment. Please see our answer to your comment to line 114.

L 243 – Add “or below” after “at”.

L 268 – Move “first” to after “distribution”.

L 277 – Change “and” to “or”.

L 278 – Add “were found either” after the parentheses.

ANSWER: Thank you for these three suggestions, we made these changes.

L 280 – Please remind the reader here that the scattered distribution had more preferred food.

ANSWER: Thank you, this has been specified (line 544) “As previously, in this condition in which the most preferred food was in the scattered distribution, ...”.

L 293 – Begin a new paragraph here.

ANSWER: Thank you, this has been done.

L 300 – Please add more information as to how competitors affected foraging decisions.

ANSWER: Competitors can influence foraging decisions by making subjects changing their direction; however, after running again our models checking with multiple testing, we did not find anymore a significant effect of the rank in any model: line 571 “Although species had no effect in the presented model ($\chi^2=3.200$, $df=2$, $p=0.202$), the analysis revealed potentially significant effect of species in paired interaction with task/season ($\chi^2=7.155$, $df=2$, $p=0.028$), the number of competitors at the box ($\chi^2=9.096$, $df=2$, $p=0.011$) and the distribution of the box ($\chi^2=8.429$, $df=2$, $p=0.015$), respectively. However, the control procedure for maintaining the false discovery rate at 5% (see paragraph “Models’ description”) revealed that those aforementioned paired interactions are likely false positive, therefore they were discarded from the model and we considered independent variables for interpretability and limitation of overparameterisation [52]”, and line 613 “Rank did not affect the overall visiting “error rate” of baited boxes ($\chi^2=2.183$, $df=1$, $p=0.139$)”,.

L 302 – Please explain how long-tailed macaques were more impacted than the other two species.

ANSWER: As explained in the previous answer, this result changed after controlling for multiple testing.

L 312 – Delete “high”.

L 313 – Change “closed” to “close”.

L 315 – Pluralize “prediction”.

L 317 – Add “a” after “had”.

L 320 – Change “removed” to “remove”.

L 327 – Change the comma to a period.

ANSWER: Thank you for these suggestions, all those six changes were made.

L 330-338 – I don’t think that it has been shown that the clumped distribution was actually chosen more.

ANSWER: We answered to this main issue of yours in answer to your comment “My main issue with the interpretation of the Results is that I am not convinced that the monkeys actually preferred the clumped distribution...” at the beginning of the file.

L 342 –Add a comma after “bias”.

L 344 – “crucial” seems like an overstatement.

ANSWER: Thank you, a comma was added and “crucial” was replaced by “important”, line 648.

L 348 – Add an “a” after “in”.

L 351 – Change “maximize” to “maximizing”.

L 352 – Change “in” to “to the”.

L 355 – “a prior” should be “a priori” in italics.

L 363 – Change “less” to “least”.

L 365 – Add “a” after “possess”.

L 369 – Add “better” before “large”.

ANSWER: These seven changes have been made.

L 370 – Change “but not a” to “compared to a”.

L 373 – Change the end of the sentence to, “render it unlikely that this species lacks small-scale spatial memory”.

L 381 – Delete the third “the”.

L 384 – Delete “also”.

ANSWER: Thank you, these four suggestions were taken into account.

L 385-386 – This sentence on humans should be deleted. It doesn’t add anything and seems tacked on.

ANSWER: We removed it.

L 390-391 – This sentence needs more explanation.

ANSWER: Thanks for this comment, we changed the sentence and we hope it is now clearer: “share a common representational format in the spatial configuration and are strongly interconnected in the mind circuits”, line 747.

L 392 – Change “Tonkeana” to “Tonkean macaques”.

L 394 – Change “strategy” to “strategies”.

ANSWER: These were fixed.

L 395 – Start a new paragraph here.

L 398 – Change “from” to “is also present in”.

L 399 – Add “the” after “influence”.

L 400 – Change “to mention” to “mentioning”.

ANSWER: Thank you for pointing out these four points. We carefully modified them.

L 411-414 – I don't really see how this relates to conservation strategies. This line is simplistic and should probably be deleted.

ANSWER: We deleted this sentence from the manuscript.

Appendix C

Dear Editor-in-Chief and Referees,

Thank you for your kind letter of “Decision on Manuscript ID RSOS-181722.R1” entitled “Where and What? Frugivory is associated with more efficient foraging in three semi-free ranging primate species” on the 28th March 2019. We made the minor revisions considering all suggestions of the reviewer and we thank him/her very much for the careful work. Please find below our specific answers to the reviewer.

Reviewer: 2

Comments to the Author(s)

The authors have done an excellent job of addressing most of the issues I had with the first draft. The manuscript, and what exactly was done, is now much clearer. Given the new models they included, I am now convinced that the monkeys preferred to forage in the clumped distribution, even though they did not choose it initially significantly more often. A bit of work is still needed to better define the predictions (see below) and clean up the English but after that it should be a nice addition to the literature on decision-making during foraging.

ANSWER: Thank you very much for this appreciation; we are pleased to know that the manuscript is now much clearer.

L 49 – Delete “a”, since the predictability is more likely annual rather than once in a lifetime.
L 72 – Reverse “feeding” and “past” to read “past feeding”.

ANSWER: Thank you, we changed both points as suggested.

L 91 – I do not think “scattered” should be used here to refer to fruit trees given how it was used in the previous sentence and in the rest of the predictions. Perhaps this is better “Since fruit trees provide ephemeral foods located in variable distributions,”.

ANSWER: Thank you for pointing out that, the sentence has been changed as suggested.

L 94 – Please define what you mean by “more affected”. Exactly how was their behavior expected to change?

ANSWER: Thank you for this comment. We meant that more frugivorous species would consider both food distribution and food quality while foraging. It is now better specified in the text (lines 103-105: “the more frugivorous primates, the *Macaca* spp. would take more into account both food quality and food distribution in comparison to the more generalist (omnivorous/insectivorous) species *S. apella*”.

L 95 – Change “do” to “did”.

L 104 – Add “The” before “Sapajus” and change “like” to “likely”.

ANSWER: This sentence has been removed since it has been confirmed that the species is *S. apella* (while the subspecies is unknown).

L 107 – Delete “N=14” here because this not your sample size per species.

L 107 – Change “fully” to “full”.

L 114 – Change “primate” to “subject”.

L 114-115 – Change “too large intakes” to “excessive intake”.

ANSWER: Thank you, we changed all six points.

L 119 – The beginning of this sentence is poorly-written. Change to “Since the three tested species are mainly frugivorous...”.

ANSWER: Thank you for this comment, the sentence has been modified as suggested.

L 128 – Add a comma after “test”.

L 151 – Delete the “to”.

L 159 – Change “that were not any more available” to “, which were no long available”.

L 163 – Change “to” to “in”.

L 173 – Change “on” to “as to”.

L 179 – Delete “on”.

L 192 – Change “because of” to “due to”.

L 193 – Change “have been” to “were”.

L 205 – Change “the each other position” to “one another”.

L 256 – Change “fist” to “first”.

L 258 – Change “have” to “had”.

L 269 – Change “Since we” to “but”.

L 277 – Change “tend” to “tended”.

L 278 – Change “forage on” to “were foraging in”.

L 280 – Change “previous” to “previously”.

L 284 – Change “fine” to “fine-scale”.

L 286 – Delete “than”.

L 300-301 – Change “despotic degree” to “degree of despotism” and “potentially ultimately affecting” to “which could potentially affect their”.

L 325 – Add “that” after “hypothesized”.

L 326 – Change “being” to “may be”.

L 328 – Change “if the variable species were as” to “whether each species was equally”.

L 329 – Change “Like so” to “Thus”.

L 338 – Change “were the individuals” to “they were”.

L 340 – Change “had not” to “did not have the”.

ANSWER: Thank you for all these suggestions, we changed the text accordingly.

L 342 – The “it” here, or what is being tested, needs to be repeated for clarity.

ANSWER: Thank you for pointing out that, the sentence has been changed: line 395, “For such a behaviour, we therefore no longer discarded unilateralised individuals, or individuals that

did not have the fruit preference matching the group preference, as long as we tested for the descriptive variables of those features (i.e., task/season and distribution of the box), while accounting for individual differences with a random factor.” We hope that now it is clearer.

L 350 – The reference needs a closing parentheses.

L 351 – I think “necessaries” needs to be changed to “necessary” here.

L 353 – Change “variables” to “variable”.

L 361-362 – Change the end of this sentence to “sequentially removing non-significant terms of higher complexity”.

L 365 – Add “a” after “we”.

L 371- Pluralize “effect”.

ANSWER: Thank you for these suggestions, we made these seven changes.

L 375 – Change again to “sequentially removing non-significant terms of higher complexity”.

ANSWER: This sentence has been removed because we already explained this procedure above (line 425: “Results displayed are only the latest version of models obtained after sequentially removing non-significant terms of higher complexity (to lower risks of overparameterisation and misinterpretation, discussed in [52] and ESM “Model selection”).

L 379 – Here there is a double-negative. Do you mean “0 for no competitor present and 1 if at least one competitor was present”?

ANSWER: We apologize for this confusion. Yes, we mean 0 for no competitor and 1 if at least one competitor was present. The sentence has been modified in this way: “except doing only a binomial variable being null if no competitor, or one if at least one competitor was present”, line 442.

L 380 – Pluralize “integer”.

L 384 – Deleted “deeply”.

L 385 – Change the middle of this sentence to, “did not seem to suffer from the reduced sample”.

L 388 – Delete “the needed”.

ANSWER: Thank you for these comments. These four changes have been made.

L 396 and 397 – I’m not sure I understand what “handed on” means in this context.

ANSWER: Thank you for pointing out this, we modified the sentence in this way: “That is, we did so for the lateralisation investigation, taking into account p-values obtained with the binomial test, for each generalised linear mixed model, using the p-values obtained with the drop1 function, and finally, for the pairwise comparisons depicted in each figure”, line 465-468. We hope that now it is clearer.

L 408 – Change “of” to “from”.

L 455 – Change “to notice” to “noticing”.

L 471 – Change “when having faced” to “after choosing”.

ANSWER: Thank you, these three changes were made.

L 491 – The way this is written with the colon and the “they” reads as if *M. tonkeana* visited non-baited boxes 5-times more, which can’t be right...?

ANSWER: Thank you for pointing out this confusion. The sentence has been modified in this way: “In terms of species, *M. tonkeana* showed significantly more goal-directed movements while visiting baited boxes (with an “error rate”, i.e. visit of non-baited boxes, close to zero according to the model predictions) compared to *S. apella*. *S. apella* visited non-baited boxes around five times more than *M. tonkeana* when following model predictions, while *M. fascicularis* significantly differed from the two aforementioned species with an intermediate error rate”, lines 575-580.

L 504 – This would read better if a comma was put after “study” and “the” was deleted.

L 508-509 – Delete “significantly more” and put “significantly more often” after “site”.

L 510 – Change “with” to “there was”.

L 512 – Add “rather” after “clumped”.

ANSWER: Thank you, we changed all four points.

L 514 and 558-561 – These results bring to mind a new paper on foraging site selection in lemurs which also found that more frugivorous species were more goal-directed and more insectivorous species are more exploratory, which is in line with your findings (Teichroeb & Vining, 2019, Navigation strategies in three nocturnal lemur species: Diet predicts heuristic use and degree of exploratory behavior. Animal Cognition).

ANSWER: Thanks a lot for suggesting this interesting article. We added a sentence at line 664: “The most likely explanation is that being more omnivorous/insectivorous than the other two study species (Table S1), they rely more on mobile resources (e.g. insects) thus revisiting potential feeding sites may be more profitable for them. Similarly, more frugivorous lemurs shown more goal-directed travels while more insectivorous lemurs shown a more exploratory behaviour”.

L 516 – Change “when” to “after”.

L 523 – Change “showed” to “shown” and “travelling cost” to “travel costs”.

L 528 – Add “and” before “planning”.

ANSWER: Thank you for these suggestions, we made all three changes.

L 562 – Please define what you mean by “opportunistic behavior” here.

ANSWER: Thanks for pointing out this. We speak about “opportunistic behaviour” when subjects visited also other non-baited feeding sites. We now better specify this in the text “Moreover, the probability that individuals adopted such an “opportunistic behaviour” (i.e.

visiting also other non-baited feeding sites) was higher for all three species (but only in two of the three tasks) when they fed in the scattered distribution.”, line 666.

L 566 – Change “have” to “had a”.

L 567-568 – Change “may pinpoint” to “suggest”.

L 599 – Changed “faced” to “chose” and “change more” to “changed the distribution they were foraging in more”.

L 603 – Add “in” after “help”.

L 608 – Add “at” after “fed”.

L 616 – Change “pursuit the” to “pursue”.

L 617 – Add “a” after “and”.

ANSWER: Thank you, we changed these seven points as suggested.